# Decoding locomotion from population neural activity in moving *C. elegans*

Kelsey M Hallinen[1†], Ross Dempsey[1†], Monika Scholz[1†‡], Xinwei Yu[1], Ashley Linder[2], Francesco Randi[1], Anuj K Sharma[1], Joshua W Shaevitz[1,3], Andrew M Leifer[1,2]*

[1]Department of Physics, Princeton University, Princeton, United States; [2]Princeton Neuroscience Institute, Princeton University, Princeton, United States; [3]Lewis-Sigler Institute of Integrative Genomics, Princeton University, Princeton, United States

**Abstract** We investigated the neural representation of locomotion in the nematode *C. elegans* by recording population calcium activity during movement. We report that population activity more accurately decodes locomotion than any single neuron. Relevant signals are distributed across neurons with diverse tunings to locomotion. Two largely distinct subpopulations are informative for decoding velocity and curvature, and different neurons' activities contribute features relevant for different aspects of a behavior or different instances of a behavioral motif. To validate our measurements, we labeled neurons AVAL and AVAR and found that their activity exhibited expected transients during backward locomotion. Finally, we compared population activity during movement and immobilization. Immobilization alters the correlation structure of neural activity and its dynamics. Some neurons positively correlated with AVA during movement become negatively correlated during immobilization and vice versa. This work provides needed experimental measurements that inform and constrain ongoing efforts to understand population dynamics underlying locomotion in *C. elegans*.

**\*For correspondence:**
leifer@princeton.edu

†These authors contributed equally to this work

**Present address:** ‡Max Planck Research Group Neural Information Flow, Center of Advanced European Studies and Research (caesar), Bonn, Germany

**Competing interests:** The authors declare that no competing interests exist.

## Introduction

Patterns of activity in an animal's brain should contain information about that animal's actions and movements. Systems neuroscience has long sought to understand how the brain represents behavior. Many of these investigations have necessarily focused on single-unit recordings of individual neurons. Such efforts have successfully revealed place cells (*O'Keefe and Dostrovsky, 1971*) and head direction cells (*Taube et al., 1990*; *Hafting et al., 2005*), for example. But there has also been a long history of seeking to understand how neural populations represent motion (*Georgopoulos et al., 1986*; *Churchland et al., 2012*; *Chen et al., 2018*). For example, population recordings from the central complex in *Drosophila* reveal that the animal's heading is represented in the population by a bump of neural activity in a ring attractor network (*Kim et al., 2017*; *Green et al., 2017*). As population and whole-brain recording methods become accessible, it has become clear that locomotory signals are more prevalent and pervasive throughout the brain than previously appreciated. For example, neural signals that correlate with rodent facial expression and body motion were recently reported in sensory areas such as visual cortex (*Stringer et al., 2019*) and in executive decision making areas of dorsal cortex (*Musall et al., 2019*).

The known locomotory circuitry in *C. elegans* focuses on a collection of pre-motor neurons and interneurons, including AVA, AVE, AVB, AIB, AIZ, RIM, RIA, RIV, RIB, and PVC that have many connections amongst themselves and send signals to downstream motor neurons involved in locomotion such as the A- or B-type or SMD motor neurons (*White et al., 1976*; *Chalfie et al., 1985*; *Zheng et al., 1999*; *Gray et al., 2005*; *Gordus et al., 2015*; *Wang et al., 2020*). These neurons can be grouped into categories that are related to forward locomotion, backward locomotion or turns.

For example, AVA, AIB, and RIM are part of a backward locomotory circuit (*Zheng et al., 1999*; *Pirri et al., 2009*; *Gordus et al., 2015*). AVB and PVC are part of a forward locomotion circuit (*Gray et al., 2005*; *Chalfie et al., 1985*; *Zheng et al., 1999*; *Li et al., 2011*; *Roberts et al., 2016*; *Xu et al., 2018*); and RIV, RIB, and RIA are related to turns (*Gray et al., 2005*; *Li et al., 2011*; *Wang et al., 2020*; *Hendricks et al., 2012*). Much of what we know about these neurons comes from recordings or manipulations of either single neurons at a time, or a selection of neurons simultaneously using sparse promoters (*Gray et al., 2005*; *Guo et al., 2009*; *Ben Arous et al., 2010*; *Kawano et al., 2011*; *Piggott et al., 2011*; *Gao et al., 2018*; *Wang et al., 2020*). Only recently has it been possible to record from large populations of neurons first in immobile (*Schrödel et al., 2013*; *Prevedel et al., 2014*; *Kato et al., 2015*) and then moving animals (*Nguyen et al., 2016*; *Venkatachalam et al., 2016*).

There has not yet been a systematic exploration of the types and distribution of locomotor related signals present in the neural population during movement and their tunings. So for example, it is not known whether all forward related neurons exhibit duplicate neural signals or whether a variety of distinct signals are combined. Interestingly, results from recordings in immobile animals suggest that population neural state space trajectories in a low dimensional space may encode global motor commands (*Kato et al., 2015*), but this has yet to be explored in moving animals. Despite growing interest in the role of population dynamics in the worm, their dimensionality, and their relation to behavior (*Costa et al., 2019*; *Linderman et al., 2019*; *Brennan and Proekt, 2019*; *Fieseler et al., 2020*) it is not known how locomotory related information contained at the population level compares to that contained at the level of single neurons. And importantly, current findings of population dynamics related to locomotion in *C. elegans* are from immobilized animals. While there are clear benefits in studying fictive locomotion (*Ahrens et al., 2012*; *Briggman et al., 2005*; *Kato et al., 2015*), it is not known for *C. elegans* how neural population dynamics during immobile fictive locomotion compare to population dynamics during actual movement.

In this work, we investigate neural representations of locomotion at the population level by recording whole-brain neural activity as the animal crawls on agar. We further construct a decoder to predict the animal's current locomotion from a linear combination of neural activity alone. The performance of the decoder gives us confidence in our ability to find locomotory signals, and allows us to study how those signals are distributed and represented in the brain.

We show that distinct subpopulations of neurons encode velocity and body curvature, and that these populations include neurons with varied tuning. We also find that the decoder relies on different neurons to contribute crucial information at different times. Finally, we compared brain-wide neural activity during movement and immobilization and observe that immobilization alters the correlation structure of neural dynamics.

## Results

To investigate locomotory-related signals in the brain, we simultaneously recorded calcium activity from the majority of the 188 neurons in the head of *C. elegans* as the animal moved, *Figure 1a–c*, (*Nguyen et al., 2016*). The animal expressed the calcium indicator GCaMP6s and a fluorescent protein RFP in the nuclei of all neurons (strain AML310).

We report calcium activity as a motion-corrected fluorescence intensity $F_{mc}$, described in methods. We measured two features of locomotion: velocity and body curvature. Velocity is computed from the movement of a point on the head of the worm as described in the methods. Body curvature is calculated as the mean curvature along the animal's centerline and has large deviations from zero during turning or coiling.

We found multiple neurons with calcium activity significantly tuned to either velocity or curvature (*Figure 1*). Some neurons were more active during forward locomotion while others were more active during backward locomotion (*Figure 1d* and *Figure 1—figure supplement 1*). Similarly some neurons were active during dorsal bends and others during ventral bends (*Figure 1e* and *Figure 1—figure supplement 2*). In some cases, the derivative of the activity was also significantly correlated with features of locomotion. We recorded from additional animals for a total of 11 animals expressing GCaMP6s (strain AML310 or AML32) and 11 control animals expressing GFP (strain AML18) and tabulated the number of significantly tuned neurons in each recording, *Figure 1—figure supplement 3*. To be classified as 'significantly tuned' the neuron's Pearson's correlation coefficient had to

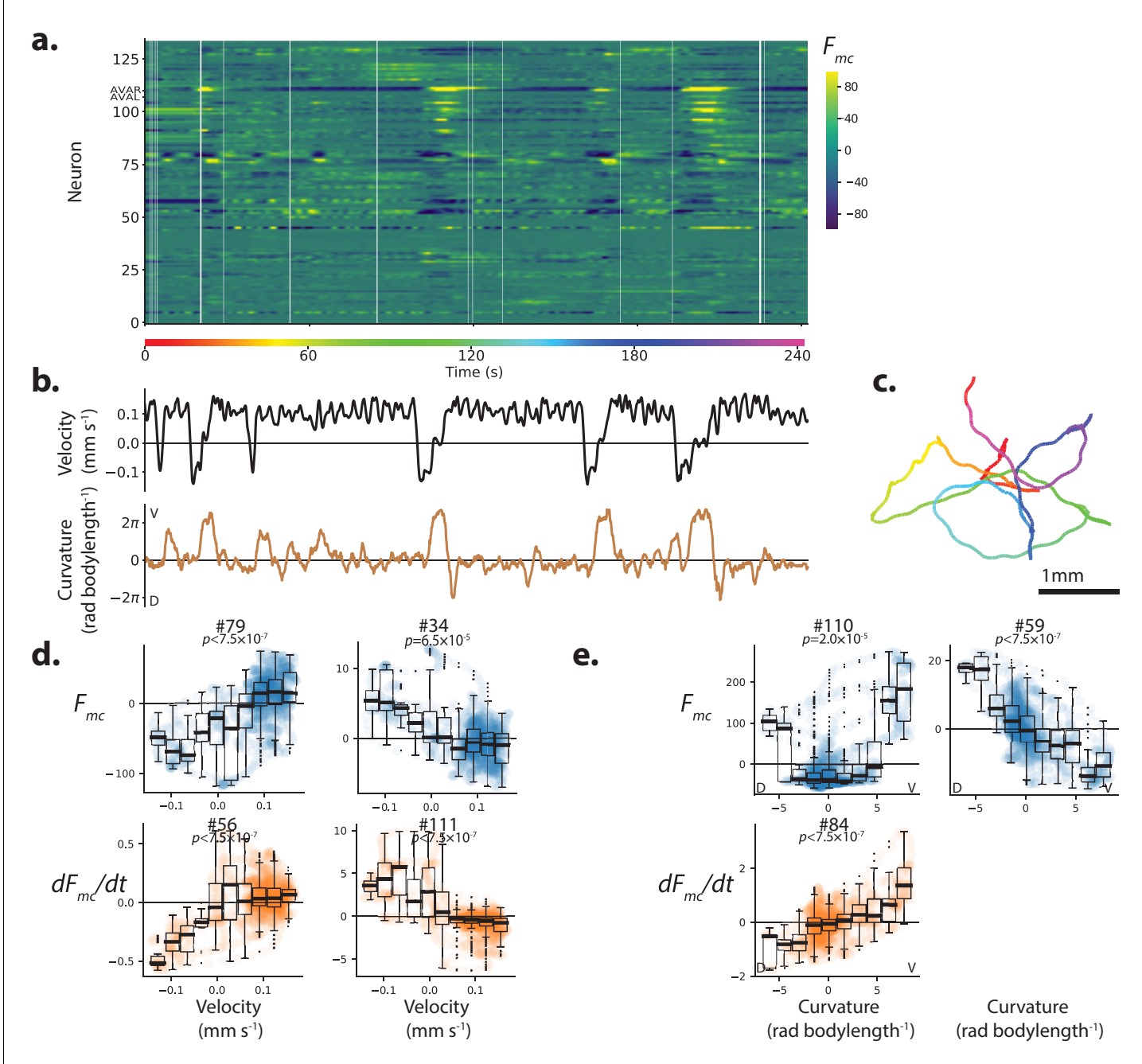

**Figure 1.** Population calcium activity and tuning of select neurons during spontaneous animal movement. Recording AML310_A. (**a**) Calcium activity of 134 neurons is simultaneously recorded during locomotion. Activity is displayed as motion-corrected fluorescent intensity $F_{mc}$. Neurons are numbered according to agglomerative hierarchical clustering. White space indicates time-points where neural tracking failed. (**b**) Body bend velocity and body curvature derived from an eigenvalue decomposition, and (**c**) position on the plate during recording are shown. (**d**) Example neurons significantly tuned to velocity. Examples are those with the highest Pearson's correlation coefficient in each category: activity (or its derivative) with positive (or negative) correlation to velocity. P-values are derived from a shuffling procedure that preserves per-neuron correlation structure. All tuning curves shown are significant at 0.05% after Bonferroni correction for multiple hypothesis testing ($p<1.9 \times 10^{-4}$). Boxplot shows median and interquartile range. Blue or orange shaded circles show neural activity at each time point during behavior. (**e**) Example neurons highly tuned to curvature were selected similarly. No neurons with negative $dF/dt$ tuning to curvature passed our significance threshold.

The online version of this article includes the following figure supplement(s) for figure 1:

**Figure supplement 1.** Additional details and examples of velocity tuning.

**Figure supplement 2.** Additional details and examples of curvature tuning.

**Figure supplement 3.** Number of significantly tuned neurons across recordings.

both pass a multiple-hypothesis corrected statistical test based on a recording-specific shuffle (described in the Materials and methods), and exceed a minimum absolute value of 0.4. The existence of neural signals correlated with these behaviors is broadly consistent with single-unit or sparse recordings during forward and backward locomotion (*Ben Arous et al., 2010*; *Kawano et al., 2011*; *Gordus et al., 2015*; *Shipley et al., 2014*; *Kato et al., 2015*; *Wang et al., 2020*) and turning (*Kocabas et al., 2012*; *Donnelly et al., 2013*; *Shen et al., 2016*; *Wang et al., 2020*).

To validate our population recordings, we investigated the well-characterized neuron pair AVAL and AVAR. We labeled those neurons using blue fluorescent protein (BFP) which is spectrally separated from the other two colors we use for neuron localization and activity (strain AML310), see *Figure 2a*. These two neurons, called AVA, are a bilaterally symmetric pair with gap junctions between them that have been shown to exhibit large calcium transients that begin with the onset of backward locomotion, peak around the end of backward locomotion during the onset of forward locomotion, and then slowly decay (*Ben Arous et al., 2010*; *Kawano et al., 2011*; *Faumont et al., 2011*; *Shipley et al., 2014*; *Gordus et al., 2015*; *Kato et al., 2015*). Our measure of AVA's activity, recorded simultaneously with 131 other neurons during movement, is consistent with prior recordings where AVA was recorded alone. We note that single-unit recordings of AVA used in previous studies lacked the optical sectioning needed to resolve these neurons separately. Here we resolve both AVAL and AVAR and find that their activities are similar to one another, and they both exhibit the expected transients timed to backward locomotion, *Figure 2b*. Signal-to-noise in AVAR is higher than AVAL because in this recording AVAR lies closer to the imaging objective lens, while AVAL is on the opposite side of the head and therefore must be imaged through the rest of the brain. We also report the sum of the individual traces in *Figure 2—figure supplement 1*. The similarity we observe between activities of AVAL and AVAR, and the similarities between our recordings of AVA and those previously reported in the literature serves to validate our ability to simultaneously record neural activity accurately from across the brain. It also suggests that the noise in this recording is modest compared to the features of interest in AVA's calcium transients.

We recorded from three additional animals and identified AVA neurons in each. The temporal derivative of AVA's activity has previously been shown to correlate with velocity over the range of negative (but not positive) velocities (*Kato et al., 2015*). Consistent with these reports, the derivative of AVA's activity, $dF_{\mathrm{mc}}/dt$, aggregated across the four population recordings has a negative correlation to velocity over the range of negative velocities, *Figure 2c*.

In our exemplar recording, AVA's activity (not its temporal derivative) also correlates with body curvature (*Figure 1e*, neuron #110). Correlation to curvature likely arises because our exemplar recording includes many long reversals culminating in deep ventral bends called 'omega turns,' that coincide in time with AVA's peak activity. Taken together, AVA's activity simultaneously recorded from the population is in agreement with prior reports where AVA activity was recorded alone.

## Population decoder outperforms best single neuron

AVA's activity is related to the animal's velocity, but its activity alone is insufficient to robustly decode velocity. For example, AVA is informative during backward locomotion, but contains little information about velocity during forward locomotion, *Figure 2c*. To gain reliable information about velocity, the nervous system will need more than the information contained in the activity of AVA. In primate motor cortex, for example, linear combinations of activity from the neural population provides more information about the direction of a monkey's arm motion during a reach task than a single neuron (*Georgopoulos et al., 1986*). In *C. elegans*, recordings from single or sparse sets of neurons show that multiple neurons have activity related to the animal's velocity or curvature (*Ben Arous et al., 2010*; *Kocabas et al., 2012*; *Kawano et al., 2011*; *Piggott et al., 2011*; *Gordus et al., 2015*; *Kato et al., 2015*; *Wang et al., 2020*). Recordings from immobilized animals further suggest that population neural dynamics in a simple low dimensional space may represent locomotion (*Kato et al., 2015*), but this has yet to be explored in moving animals.

We sought to explicitly compare the information about velocity and curvature contained in the population to that contained in a single neuron. To access information in the population, we constructed a decoder that uses linear regression with regularization to relate the weighted sum of neurons' activity to either velocity or curvature. Ridge regression (*Hoerl and Kennard, 1970*) was performed on 60% of the recording (training set) and the decoder was evaluated on a held-out test-

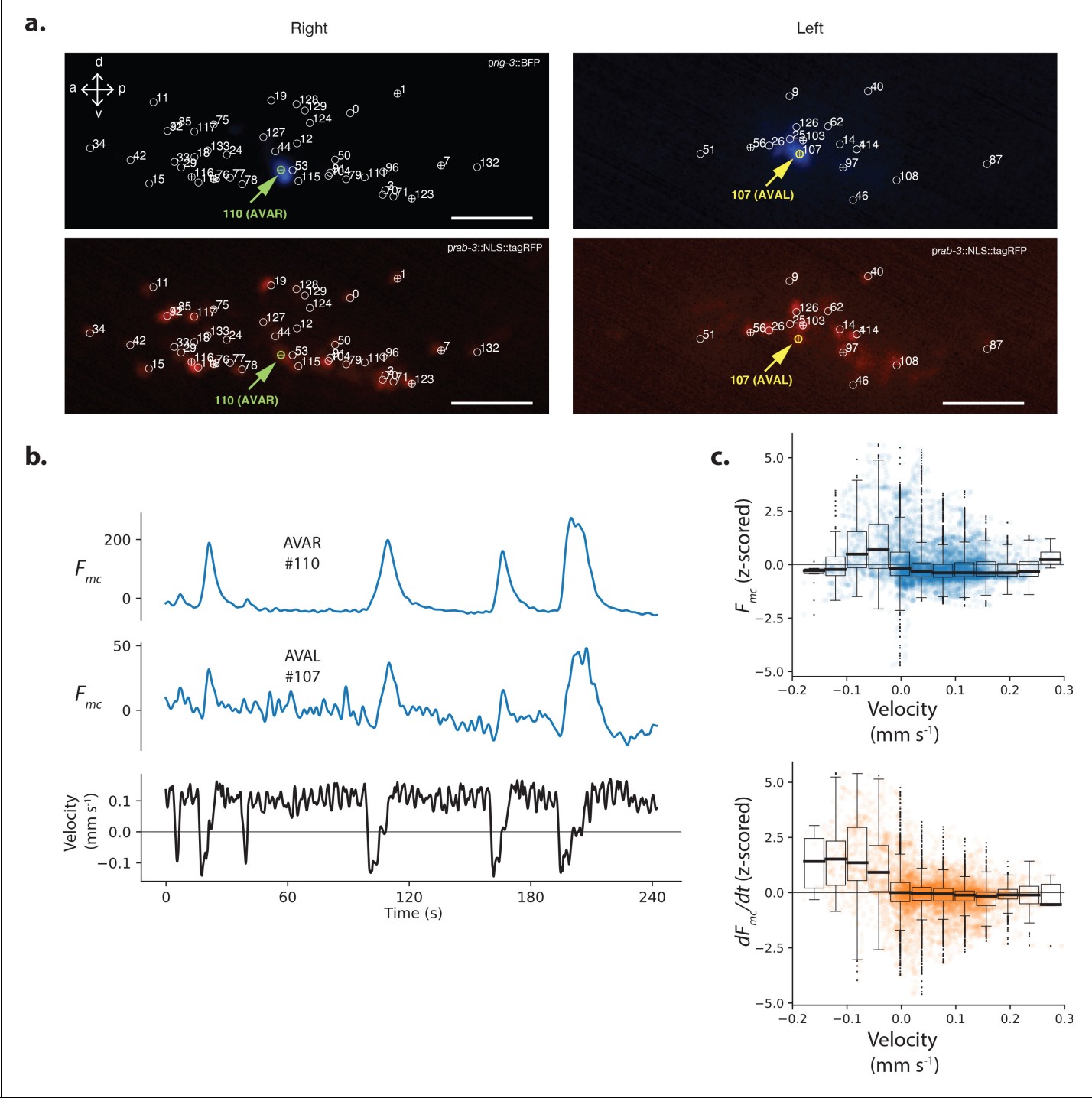

**Figure 2.** Neuron pair AVA is active during backward locomotion and exhibits expected tuning during moving population recordings. (**a**) AVAR and AVAL are labeled by BFP under a *rig-3* promoter in strain AML310. Two optical planes are shown from a single volume recorded during movement. Planes are near the top and bottom of the optical stack, corresponding to the animals' extreme right and left. The recording is the same as in *Figure 1*. Top row shows BFP. Bottom row shows RFP in the nuclei of all neurons. Segmented neurons centered in the optical plane are labeled with ⊕, while neurons from nearby optical planes are labeled with ○. Arrow indicates AVAR or AVAL. Numbering corresponds to *Figure 1a*. (**b**) Calcium activity of AVAR and AVAL during locomotion in recording AML310_A, same as in *Figure 1*. (**c**) Aggregate tuning of AVA across four individuals (seven neurons). Boxplot shows median and interquartile range. Lightly shaded blue or orange circles show activity at each time point during behavior.

The online version of this article includes the following figure supplement(s) for figure 2:

**Figure supplement 1.** Sum of AVAL and AVAR activity.

set made up of the remaining 40% of the recording (shaded green in *Figure 3a,c*). Evaluating performance on held-out data mitigates potential concerns that performance gains merely reflect over-fitting. In the context of held-out data, models with more parameters, even those that are over-fit, will not inherently perform better. Cross-validation was used to set hyper-parameters. Two regression coefficients are assigned to each neuron, one weight for activity and one for its temporal derivative. We compared performance of the population decoder on the held-out test set to that of the most

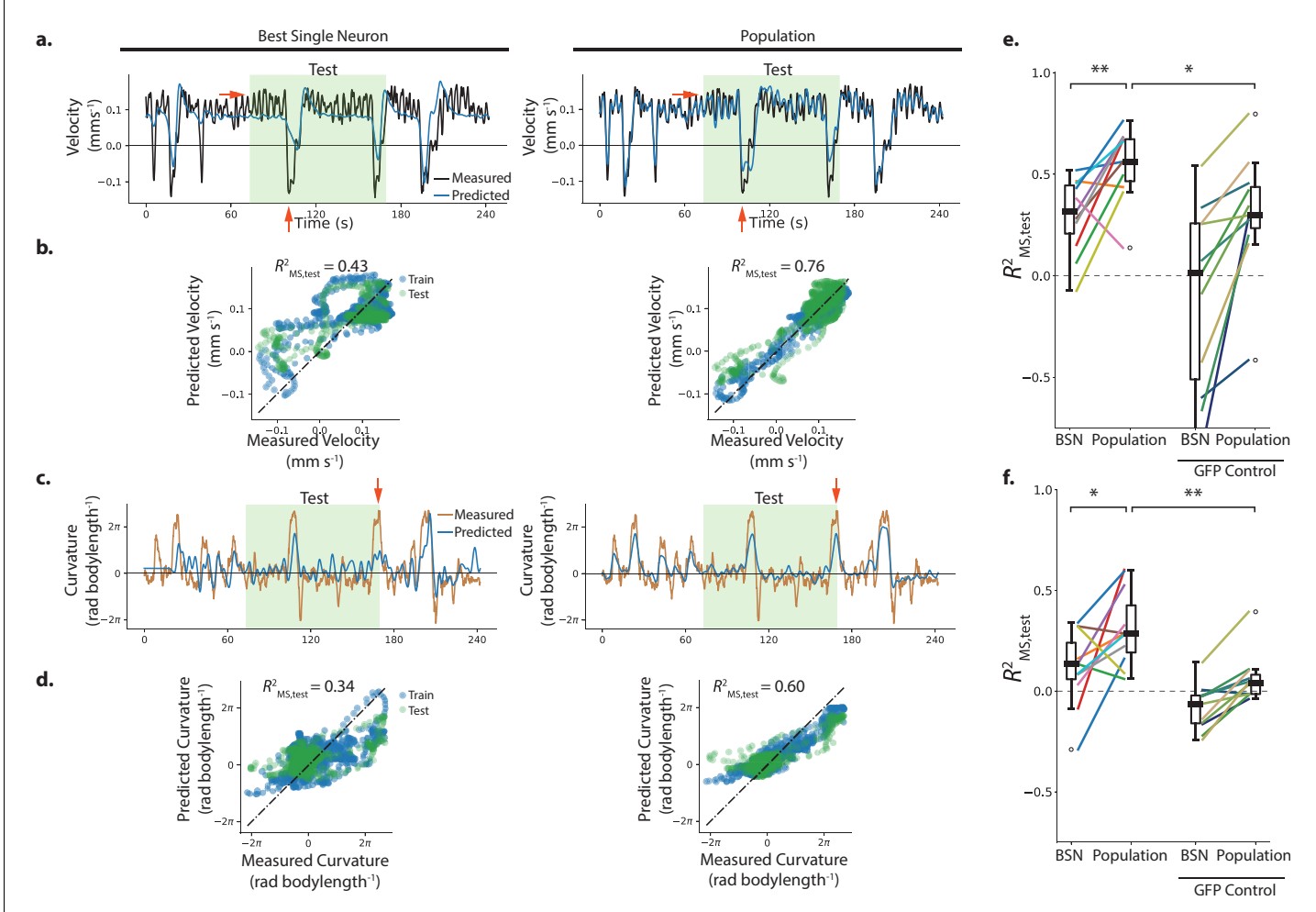

**Figure 3.** Population neural activity decodes locomotion. (a–d) Performance of the best single neuron (BSN) is compared to a linear population model in decoding velocity and body curvature for the exemplar recording AML310_A shown in *Figure 1*. (a) Predictions on the held-out test set are compared to measured velocity. Light green shaded region indicates held-out test set. Red arrows indicate examples of features that the population captures better than the BSN. (b) Performance is reported as a coefficient of determination $R^2_{MS}$ evaluated on the mean-subtracted held-out test data (green points). (c,d) Model predictions are compared to measured curvature. (e) Performance of velocity decoding is shown for recordings of $n = 11$ individuals (strain AML310 and AML32) and for recordings of $n = 11$ GFP control animals lacking a calcium indicator (strain AML18). Two-sided Wilcoxon rank test is used to test significance of population performance compared to BSN, $p = 3.9 \times 10^{-3}$. Welch's unequal variance t-test is used to test significance of population performance compared to GFP control, $p = 3.2 \times 10^{-2}$. (f) Performance of curvature decoding is shown for all recordings. Each recording is colored the same as in e. $p = 3.2 \times 10^{-2}$ and $p = 1.8 \times 10^{-3}$ for comparisons of population performance to that of BSN, and GFP control, respectively.

The online version of this article includes the following figure supplement(s) for figure 3:

**Figure supplement 1.** Performance correlates with maximal GCaMP Fano Factor, a metric of signal.

**Figure supplement 2.** Neural activity and behavior for all moving calcium imaging recordings (AML310 and AML32).

**Figure supplement 3.** Neural activity and behavior for all moving GFP control recordings (AML18).

**Figure supplement 4.** Alternative population models.

**Figure supplement 5.** Nonlinear fits using best single neuron.

correlated single neuron or its derivative on the same held-out test set. Performance is reported as a coefficient of determination on the mean-subtracted held out test set $R^2_{\mathrm{ms,test}}$.

For the exemplar recording shown in *Figure 1* and *Figure 2a–b*, the population performed better on the held-out-test set than the most correlated single neuron (or its temporal derivative) for both velocity and body curvature, see *Figure 3*. For velocity, population performance was $R^2_{\mathrm{ms,test}} = 0.76$ compared to $R^2_{\mathrm{ms,test}} = 0.43$ for the best single neuron; and for curvature population performance was $R^2_{\mathrm{ms,test}} = 0.60$ compared to $R^2_{\mathrm{ms,test}} = 0.34$ for the best single neuron. Red arrows in *Figure 3* highlight striking behavior features that the best single neuron misses but that the population decoder captures. We also explored alternative population models, including both linear and non-linear models with different features, cost penalties, and differing number of parameters *Figure 3— figure supplement 4* (parameters described in Materials and methods and Table 5) Of the populations models we tried, the model used here was one of the simplest and also had one of the best mean performances at decoding velocity across all recordings, *Figure 3—figure supplement 4*.

Activity was recorded from a total of 11 moving animals (*Figure 3—figure supplement 2*) and the linear population model was used to decode each recording ($n = 7$ recordings of strain AML32; $n = 4$ recordings of strain AML310, also shown in *Figure 2c*). The population model was compared to the best single neuron in each recording. Because the correspondence between neurons across animals is not known in these recordings, the identities of neurons used by the population decoder and that of the specific best single neuron may vary from recording to recording. The population significantly outperformed the best single neuron at decoding the held-out portions of the recordings for both velocity and curvature (p < 0.05 two-sided Wilcoxon rank test).

There was large worm-to-worm variability in the performance of the decoders. Performance across recordings correlated with one metric of the signal in our recordings, the maximal Fano factor across neurons of the raw time-varying GCaMP fluorescence intensity,

$$\mathrm{Fano_{GCaMP}} = \max_i \left( \frac{\sigma^2[F_{i,\mathrm{GCaMP}}]}{\mu[F_{i,\mathrm{GCaMP}}]} \right), \tag{1}$$

where $max_i$ indicates the maximum over all neurons in the recording, and $\sigma^2$ and $\mu$ are the variance and mean respectively of the raw GCaMP activity of the neuron, see *Figure 3—figure supplement 1*. Here, the variance term is related to the signal in the recording. The recording with the highest $\mathrm{Fano_{GCaMP}}$ performed best at decoding velocity and curvature. This suggests that variability in performance may be due in part to variability in the amount of neural signal in our recordings.

In some recordings, where the population outperforms the best single neuron, it does so in part because the population decodes a fuller range of the animal's behavior compared to the best single neuron. Recording AML32_A shows a striking example: the best single neuron captures velocity dynamics for negative velocities, but saturates at positive velocities. The population decoder, by contrast, captures velocity dynamics during both forward and backward locomotion during the held-out test set, and covers a larger fraction of the held-out velocity range, see *Figure 4*.

Motion artifact is of potential concern because it may resemble neural signals correlated to behavior (*Nguyen et al., 2016*; *Chen et al., 2013*). For example, if a neuron is compressed during a head bend, it may increase local fluorophore density causing a calcium-independent increase in fluorescence that would erroneously appear correlated with head bends. We address this concern in all our recordings by extracting a motion corrected calcium signal derived from a comparison of GCaMP and RFP dynamics in the same neuron. All strains in this work express a calcium-insensitive RFP in every neuron in addition to GCaMP. Motion artifacts should affect both fluorophores similarly. Therefore, the motion correction algorithm subtracts off those dynamics that are common to both GCaMP and RFP timeseries (details in Materials and methods).

To validate our motion correction, and to rule out the possibility that our decoder primarily relies on non-neural signals such as those from motion artifact, we recorded from control animals lacking calcium indicators. These animals expressed GFP in place of GCaMP (11 individuals, strain AML18, RFP was also expressed in all neurons). GFP emits a similar range of wavelengths to GCaMP but is insensitive to calcium. Recordings from these control animals were subject to similar motion artifact but contained no neural activity because they lack calcium sensors (*Figure 3—figure supplement 3*). Recordings from GFP control animals were subject to the same motion correction as GCaMP animals. For both velocity and curvature, the average population model performance was significantly

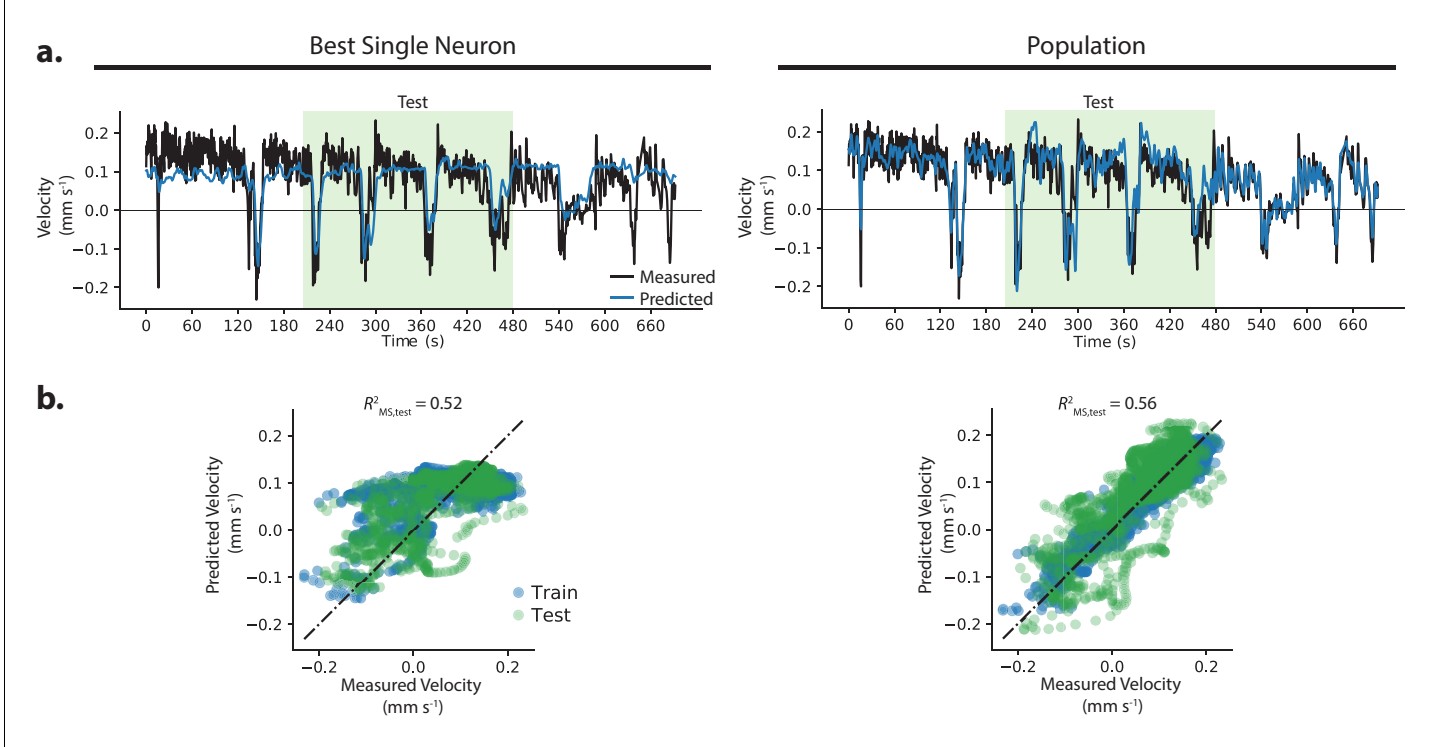

**Figure 4.** Example where population decoded a fuller range of animal behavior. (a) The decoding from the best single neuron and the population model are compared to the measured velocity for example recording AML32_A. (b) Predictions from the best single neuron saturate at a velocity of approximately 0.1 mm s$^{-1}$.

worse at decoding calcium-insensitive GFP control recordings than the calcium-sensitive GCaMP recordings (*Figure 3e–f*, median performance $R^2_{\mathrm{ms,test}} = 0.56$ for GCaMP compared to 0.30 for GFP control at decoding velocity, and median performance $R^2_{\mathrm{ms,test}} = 0.29$ for GCaMP compared to 0.04 for GFP control for curvature (p < 0.05 Welch's unequal variance test), suggesting that the decoder's performance relies on neural signals. Taken together, we find that a simple linear combination of neurons performs better at decoding velocity or curvature than the best single neuron, and that the population decoder is not primarily relying on motion artifact.

## Types of signals used to decode from the population

We further sought to understand how information across the population was utilized by the decoder. We were interested in this for two reasons, first because it should provide insights into how the population model is able to decode effectively. And second, because an effective strategy adopted by the decoder may also be available to the brain, therefore understanding how the decoder works also illustrates plausible strategies that the brain could employ to represent locomotion.

To investigate how the decoder utilizes information from the population, we inspect the neural weights assigned by the decoder. The decoder assigns one weight for each neuron's activity, $W_F$, and another for the temporal derivative of its activity, $W_{\frac{dF}{dt}}$. It uses ridge regularization to penalize weights with large amplitudes, which is equivalent to a Bayesian estimation of the weights assuming a zero-mean Gaussian prior. In the exemplar recording from *Figure 1*, the distribution of weights for both velocity and curvature are indeed both well-approximated by a Gaussian distribution centered at zero. This suggests that the decoder does not need to deviate significantly from the prior in order to perform well. In particular, although changing the sign of any weight would not incur a regularization penalty, the decoder relies roughly equally on neurons that are positively and negatively tuned to velocity, and similarly for curvature.

At the population level, the decoder assigns weights that are roughly distributed evenly between activity signals $F$ and temporal derivative of activity signals $dF/dt$ (*Figure 5a,b*). But at the level of

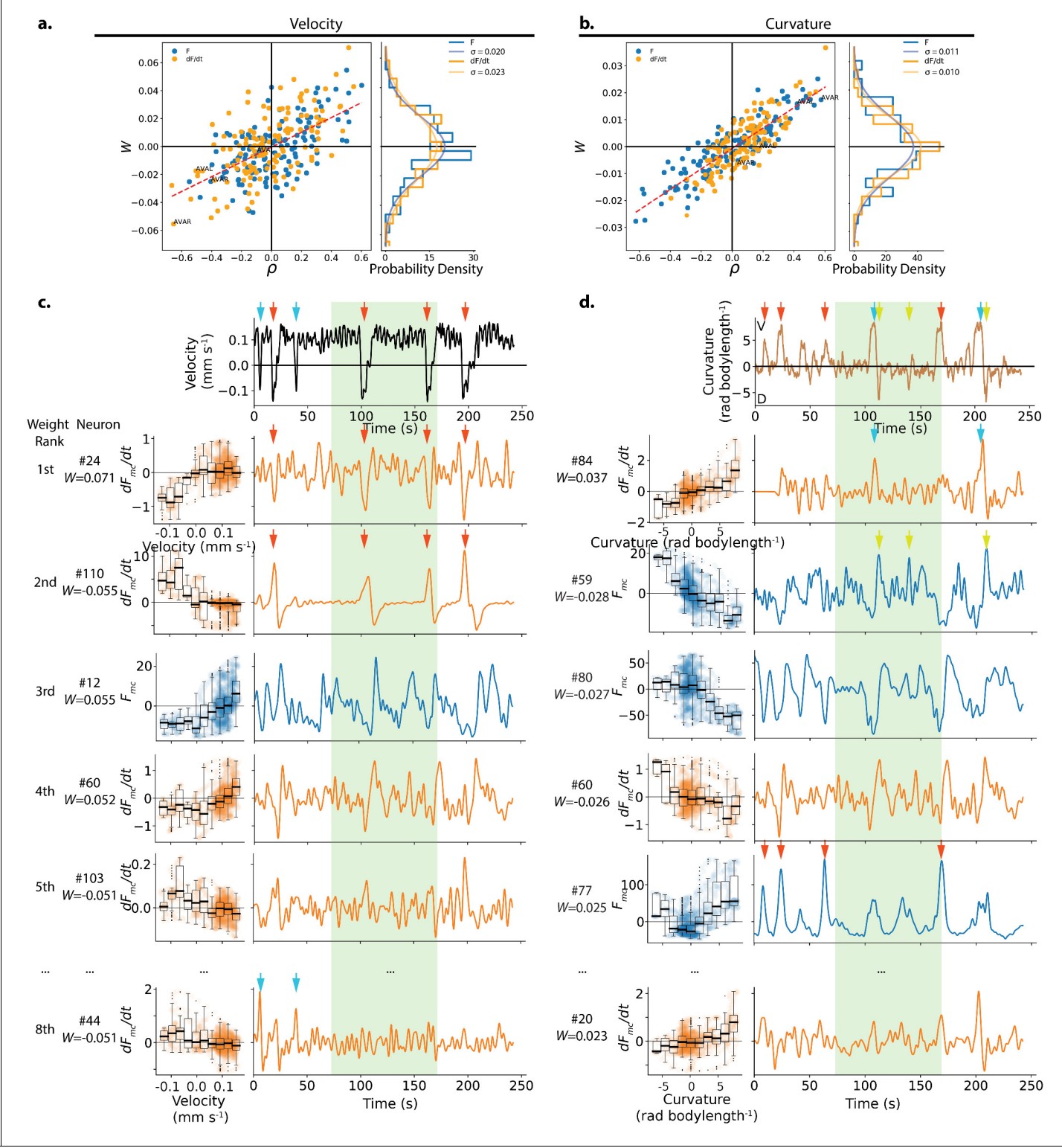

**Figure 5.** Weights assigned to neurons by the population model in the exemplar recording, and their respective tuning. (**a**) The weight $W$ assigned to each neuron's activity ($F_{mc}$) or its temporal derivative ($dF_{mc}/dt$) by the velocity population decoder is plotted against its Pearson's Correlation coefficient ρ which characterizes its tuning to velocity. Recording AML310_A is shown, same as in *Figure 1*. Dashed red line shows line of best fit. Right panel shows the observed distribution of weights. A zero-mean Gaussian with standard deviation set to the empirically observed standard deviation is also shown. (**b**) Same as in a, but for curvature. (**c**) Tuning and activity of the top highest amplitude weighted neurons for velocity is shown. Activity of each neuron is time aligned to the observed behavior (top row). Neurons are labeled corresponding to their number in the heatmap in *Figure 1*. Their rank

*Figure 5 continued on next page*

*Figure 5 continued*

and weight $W$ in the decoder is listed. Red arrows highlight peaks in the temporal derivative of activity of neuron #24 and #110, while cyan arrows highlight peaks of neuron #44. Y- and X-axes labels and scales are preserved within individual rows and columns, respectively. Light green shading indicates the held-out portion of the recording. (**d**) Same as c but for curvature. Red and cyan arrows show two sets of deep ventral bends that are captured by different neurons. Green arrows show dorsal bends.

The online version of this article includes the following figure supplement(s) for figure 5:

**Figure supplement 1.** Comparison of weights assigned to a neuron's activity versus its temporal derivative.

**Figure supplement 2.** Comparison of weights assigned for decoding velocity vs decoding curvature.

**Figure supplement 3.** Example traces of highly weighted neurons used to decode curvature in AML32_A.

individual neurons, the weight assigned to a neuron's activity $W_F$ was not correlated with the weight assigned to the temporal derivative of its activity $W_{\frac{dF}{dt}}$ (***Figure 5—figure supplement 1***). Again, this is consistent with the model's prior distribution of the weights. However, given that the model could have relied more heavily on either activity signals $F$ or on temporal derivative signals $dF/dt$ without penalty, we find it interesting that the decoder did not need to deviate from weighting them roughly equally in order to perform well.

We wondered what types of signals are combined by the decoder. For example, it is conceptually useful to consider a simple null hypothesis in which multiple neurons exhibit exact copies of the same behavior-related signal with varying levels of noise. In that case, the population decoder would outperform the best single neuron merely by summing over duplicate noisy signals. We inspected the activity traces of the top weighted neurons in our exemplar recording (***Figure 5c,d***). Some highly weighted neurons had activity traces that appeared visually similar to the animal's locomotory trace for the duration of the recording (e.g.#80 for curvature) and other neurons had activity that might plausibly be noisy copies of each other (e.g. #12 and #60 for velocity). But other highly weighted neurons had activity traces that were distinct or only matched specific features of the locomotory behavior. For example, negatively weighted neuron #59 exhibited distinct positive peaks during dorsal turns (green arrows), but did not consistently exhibit corresponding negative peaks during ventral turns. This is consistent with prior reports of neurons such as SMDD that are known to exhibit peaks during dorsal but not ventral head bends (***Hendricks et al., 2012***; ***Shen et al., 2016***; ***Kaplan et al., 2020***).

In the recording shown, we also find some neurons that have activity matched to only specific instances of a behavior motif. For example, the temporal derivative of the activity of neuron #84 contributes distinct peaks to ventral bends at approximately 105 s and 210 s, but not during similar ventral turns at other time points (***Figure 5d***, blue arrows). Conversely, highly weighted neuron #77 contributes sharp peaks corresponding to four other ventral bends (***Figure 5d***, red arrows) that are absent from neuron #84. Similarly (although perhaps less striking) for velocity, neurons #24 and #110 contribute peaks for one set of reversals (***Figure 5c***, red arrows), while neuron #44 contributes peaks to a complimentary set of two reversals (***Figure 5c***, blue arrows). Similarly in recording AML32_A, different neurons contribute peaks of activity corresponding to different sets of ventral or dorsal turns, ***Figure 5—figure supplement 3***. While we observed this effect in some recordings, it was not obviously present in every recording.

From this inspection of highly weighted neurons, we conclude that in at least some recordings the decoder is not primarily averaging over duplicate signals. Instead the decoder sums together different types of neural signals, including those that capture only a certain feature of a behavior (e.g. dorsal turns or ventral turns, but not both) or that seemingly capture only certain instance of the same behavior motif (some reversals but not others).

## Majority of decoder's performance is provided by a subset of neurons

We wondered how many neurons the model relies upon to achieve most of its performance. The magnitude of a neuron's assigned weight reflects its relative usefulness in decoding locomotion. Therefore we investigated performance of a restricted population model that had access to only those $N$ neurons that were most highly weighted by the full model. We sequentially increased the number of neurons $N$ and evaluated the partial model performance (***Figure 6—video 1***). In this way, we estimated the number of neurons needed to first achieve a given performance (***Figure 6a***).

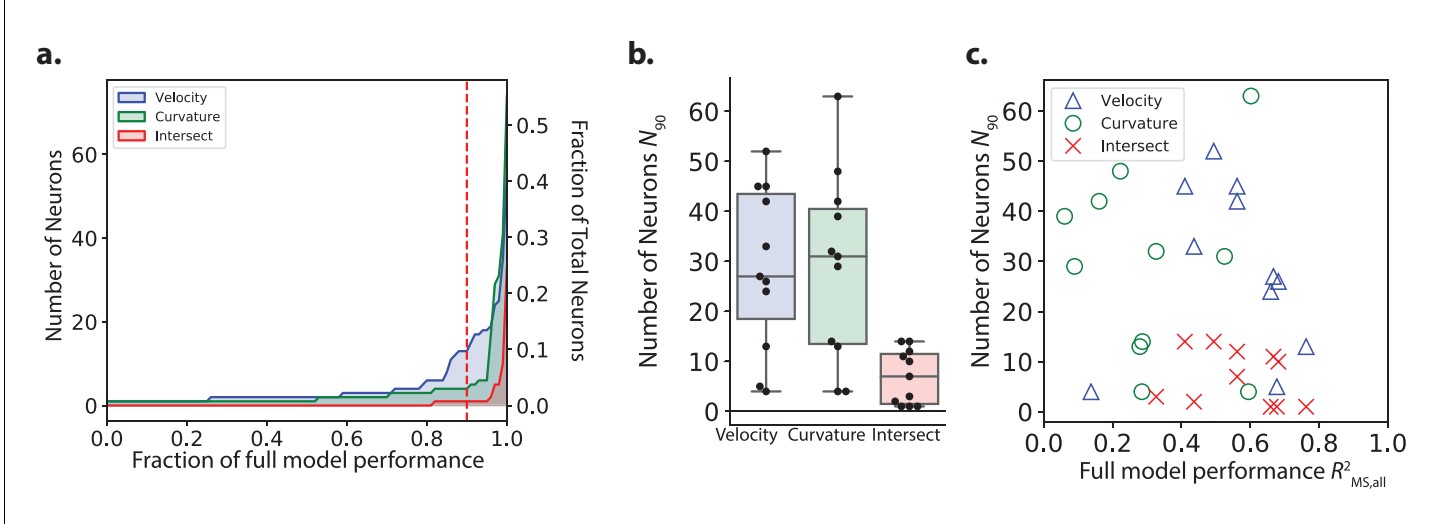

**Figure 6.** Number of neurons needed by the model to decode velocity and curvature. (**a**) The minimum number of neurons needed for a restricted model to first achieve a given performance is plotted from recording AML310_A in *Figure 1*. Performance, $R^2_{MS,all}$ is reported separately for velocity (blue) and curvature (green) and is calculated on the entire recording (test and train). Intersect refers to the intersection of the set of neurons included in both partial models (velocity and curvature) for a given performance. Red dashed line, $N_{90}$, indicates number of neurons needed to achieve 90% of full model performance. (**b**) $N_{90}$ is computed for velocity and curvature for all recordings. The number of neurons present in both populations at 90% performance level (intersection) is shown. Box shows median and interquartile range. (**c**) $N_{90}$ for all recordings is shown plotted versus the performance of the full population velocity or curvature decoder, respectively. Number of intersection neurons (red 'x') is plotted at the higher of either the velocitys or curvature's performance.

The online version of this article includes the following video for figure 6:

**Figure 6—video 1.** Animation showing partial model performance as neurons are added, corresponding to *Figure 6a*.

https://elifesciences.org/articles/66135#fig6video1

Because we were interested in probing the particular successful set of weights that the model had found, we constrained the relative weights of neurons in the partial model to match those of the full model. We note that adding a neuron gave the model access to both that neuron's activity and its temporal derivative. We define the number of neurons needed to first achieve 90% full model performance as the $N_{90}$ and use this value as an estimate of the number of important neurons for decoding. For the exemplar recording AML310_A, 90% of the model's performance was achieved when including only 13 neurons for velocity, and only four neurons for curvature.

Across all recordings, we saw large variability in the number of important neurons $N_{90}$ (*Figure 6b,c* and *Table 1*) with a median of 27 neurons for velocity and 31 for curvature. By comparison, our recordings contained a median total of 121 neurons. On average, the decoder relies on roughly a quarter of the neurons in a recording to achieve the majority of its decoding performance.

### Largely distinct sub-populations contain information for velocity and curvature

We wondered how a neuron's role in decoding velocity relates to its role in decoding curvature. Most neurons that have been well characterized in the literature, such as AVE and SMD, have been ascribed roles to either velocity or curvature but not both. RIB may be exception, and has recently been proposed to be involved in both reversals and turns (*Wang et al., 2020*). In the exemplar

**Table 1.** Number of neurons needed to achieve 90% of full model performance, $N_{90}$, reported as (median ± standard deviation), across all 11 recordings.

| Velocity $N_{90}$ | Curvature $N_{90}$ | Intersection $N_{90}$ | Total recorded |
|---|---|---|---|
| 27 ± 16 | 31 ± 18 | 7 ± 5 | 121 ± 12 |

recording AML310_A, there was no obvious population-wide trend between the magnitude of a neuron's weight at decoding velocity and the magnitude of its weight at decoding curvature for either $F$, $dF/dt$ or both, see *Figure 5—figure supplement 2*. Furthermore, only one neuron had overlap between the $N_{90} = 13$ neurons needed to achieve 90% of full model performance at decoding velocity and the $N_{90} = 4$ neurons needed for curvature in this recording, see *Figure 6a*. Across all recordings, only $7 \pm 5$ (median ± std) neurons were included in both $N_{90}$ for the velocity and curvature sub-populations, labeled 'intersect' neurons in *Figure 6b,c* and *Table 1*. Taken together, this suggests that largely distinct sub-populations of neurons in the brain contain the majority of information important for decoding velocity and curvature.

## Immobilization alters the correlation structure of neural dynamics

Recordings of brain-wide calcium activity of immobilized *C. elegans* provided evidence to suggest that the population may be involved in representing locomotion or motor commands (*Kato et al., 2015*). Specifically these motor commands may be represented as neural trajectories through a low-dimensional state space defined by principal components determined by the correlation structure of population neural activity in the recording. Those experiments also noted some differences between the activity of neurons in immobilized population recordings and the same neuron recorded alone in a moving animal. For example, neuron RIM exhibited seemingly slower dynamics in immobilized population recordings than in sparse recordings during movement. We wondered what changes may exist at the population level between moving and immobilized animals.

We recorded population activity from a moving animal crawling in a microfluidic chip and then immobilized that animal partway through the recording by delivering the paralytic levamisole, as has been used previously (*Gordus et al., 2015*; *Kato et al., 2015*). Neural dynamics from the same population of neurons in the same animal were therefore directly compared during movement and immobilization, *Figure 7*.

Immobilization changed the correlation structure of neural activity. Clusters of neurons that had been correlated with one another during movement were no longer correlated during immobilization (see *Figure 7e*, top row, blocks of contiguous yellow on the diagonal during movement that are absent or disrupted during immobilization ). Notably, many neurons that had been only weakly positively correlated or had negative correlations during movement became strongly positively correlated with one another during immobilization forming a large block (*Figure 7e*, bottom, large contiguous yellow square that appears on the lower right along the diagonal during immobilization).

To further quantify the change in correlation structure, we defined a dissimilarity metric, the root mean squared change in pairwise correlations $\sqrt{\left\langle (\rho'_{i,j} - \rho_{i,j})^2 \right\rangle}$, and applied it to the correlation matrices during movement and immobilization within this recording, and also to two additional recordings with paralytic. As a control, we also measured the change in correlation structure across two similar time windows in the 11 moving recordings. The change in correlations from movement to immobilization was significantly larger than changes observed in correlations in the moving-only recordings ($p = 1.2 \times 10^{-2}$, Welch's unequal variance t-test), *Figure 7f*. This suggests that immobilization alters the correlation structure more than would occur by chance in a moving worm.

We next inspected the neural dynamics themselves (*Figure 7a,c*). Low-dimensional stereotyped trajectories, called manifolds, have been suggested to represent *C. elegans* locomotion in a neural state-space defined by the first three principal components of the temporal derivative of neural activity (*Kato et al., 2015*). We therefore performed Principal Components Analysis (PCA) on the neural activity (or its temporal derivative) of our recording during the immobilization period, so as to generate a series of principal components or PC's that capture the major orthogonal components of the variance during immobilization. Population activity during the entire recording was then projected into these first three PCs defined during immobilization, *Figure 7c*. Neural state space trajectories during immobilization were more structured and stereotyped than during movement and exhibited similarities to previous reports, see *Figure 7c,d*.

Recordings from a second animal was similar and showed pronounced cyclic activity in the first PC of the temporal derivative of neural activity, see *Figure 7—figure supplement 1b,c*. Neural state space trajectories were even more striking and periodic in recordings where the animal had been immobilized for many minutes prior to recording (see *Figure 7—figure supplement 2*, especially

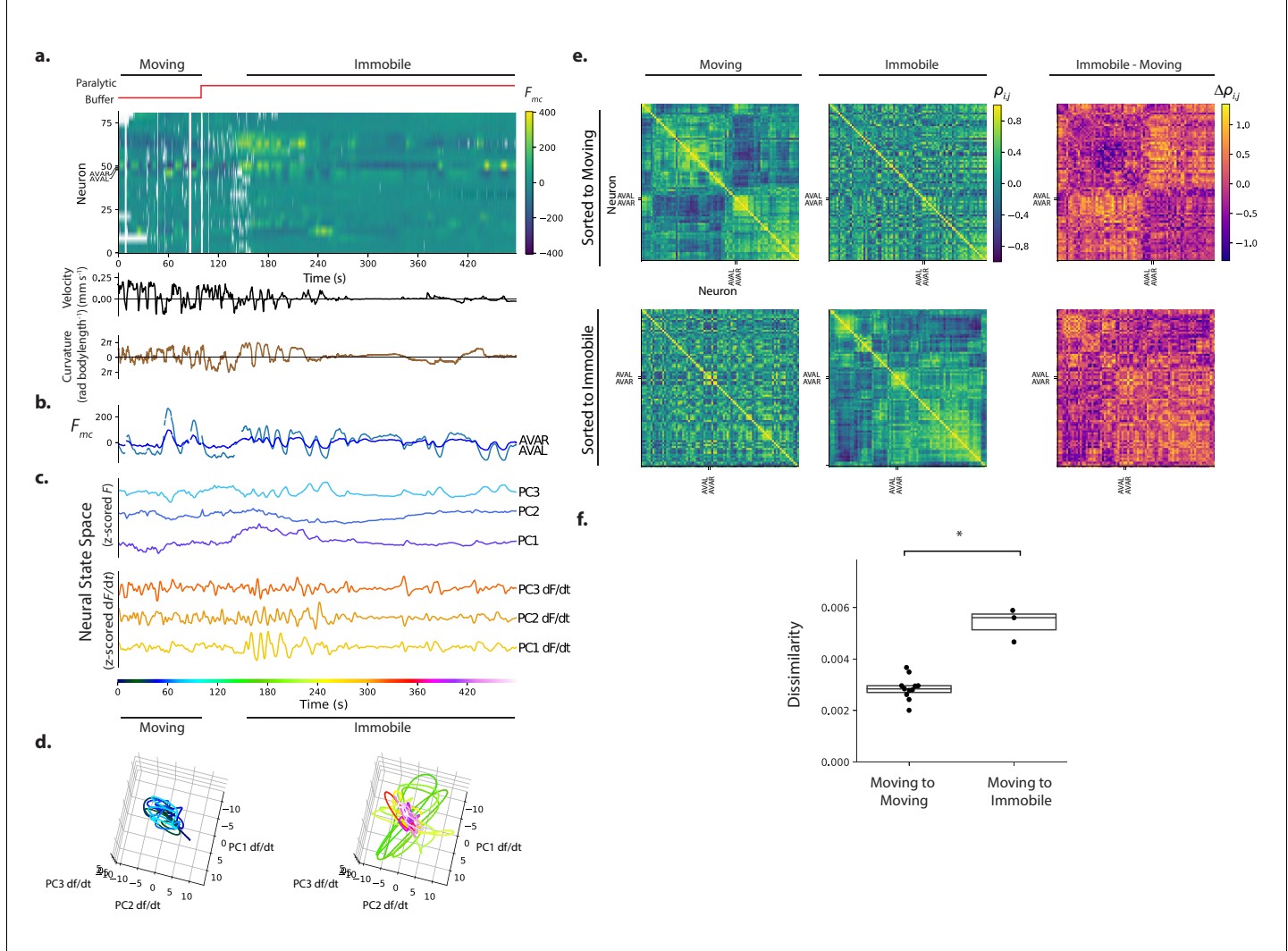

**Figure 7.** Immobilization alters the correlation structure of neural activity. (a) Calcium activity is recorded from an animal as it moves and then is immobilized with a paralytic drug, recording AML310_E. (b) Activity of AVAL and AVAR from (a). (c) Population activity (or its temporal derivative) from (a) is shown projected onto its first three PCs, as determined by only the immobilized portion of the recording. (d) Neural state space trajectories from (c) are shown split into moving and immobile portions, color coded by time. Scale, axes and PCs are the same in both plots. (e) Pairwise correlations of neural activity $\rho_{i,j}$ are shown as heatmaps for all neurons during movement and immobilization, sorted via a clustering algorithm. Top row is sorted to movement, bottom row is sorted to immobilization. (f) Dissimilarity between correlation matrices for moving and immobile portions of a recording are shown compared to the dissimilarity observed between correlation matrices taken at similar time windows within moving-only recordings. Dissimilarity is $\sqrt{\left\langle \left(\rho'_{i,j} - \rho_{i,j}\right)^2 \right\rangle}$. Dissimilarity was measured in three moving-immobile recordings with paralytic and 11 moving-only recordings. $p = 1.2 \times 10^{-2}$, Welch's unequal variance t-test. Boxes show median and interquartile range.

The online version of this article includes the following figure supplement(s) for figure 7:

**Figure supplement 1.** Example from additional moving-to-immobile recording.

**Figure supplement 2.** Immobile-only recording.

PC1). The emergence of structured neural state-space dynamics during immobilization is consistent with the significant change to the correlation structure observed in neural activity. Taken together, these measurements suggest that immobilization alters the correlation structure and dynamics of neural activity and may have implications for the interpretation of immobile neural dynamics.

We further investigated the activity of neuron pair AVA and its correlation to other neurons during movement and immobilization in the recording shown in *Figure 7*. AVA's activity was roughly consistent with prior reports. During movement AVA exhibited a sharp rise in response to most

instances of the animal's backward locomotion, as expected (*Figure 7b*). During immobilization, AVA exhibited slow cycles of activity captured in one of the first three PCs.

And during both movement and immobilization AVAL and AVAR were consistently highly correlated with one another ($\rho$>0.89) and participated in a small cluster of positively correlated neurons (most clearly visible in *Figure 7e* bottom row, small block around AVA).

Interestingly, immobilization induced many neurons to change the sign of their correlations with AVA. For example, some neurons, such as #43 and #44, that had negative correlation coefficients with respect to AVA during movement but had positive correlation coefficients during immobilization (Fig *Figure 8a,b,d*). Similarly, some neurons, such as #23 and #33 that had positive correlation coefficients with respect to AVA during movement, had negative correlation coefficients with respect to AVA during immobilization. On average, neurons in this recording become significantly more positively correlated to AVA upon immobilization than during movement (p = 0.019 Wilcoxon ranked test), *Figure 8c*.

Taken together, our measurements show that immobilization significantly alters the correlation structure of neural activity. Immobilization also causes neurons to change their correlation with known well-characterized neurons, like AVA, from negatively correlated to positively correlated, or vice versa.

## Discussion

Our measurements show that a linear decoder can predict the animal's current velocity and body curvature from neural signals in the population. This suggests that a linear combination of activity from different neurons is one plausible mechanism that the brain may employ to represent behavior. However, our results do not preclude the brain from using other methods for representing behavior. And in all cases, the measurements here do not distinguish between neural signals that drive locomotion, such as motor commands; and neural signals that monitor locomotion generated elsewhere, such as proprioceptive feedback (*Wen et al., 2012*). The decoder likely uses a mix of both. Future perturbation studies are needed to distinguish population-level signals that drive locomotion from those that monitor locomotion.

How should we interpret the finding that the decoder is linear? It has been observed that even very non-linear neural systems can encode information linearly. For example, the vertebrate retina has many highly non-linear connections but a linear decoder performs indistinguishably from an (nonlinear) artificial neural network at decoding visual signals from populations of retinal ganglion cells (*Warland et al., 1997*). *C. elegans* may be another example, like the retina, of a non-linear system that represents information linearly. The *C. elegans* nervous system, however, also contains known instances of connections that appear linear over a physiologically relevant range of activities (*Liu et al., 2009*; *Lindsay et al., 2011*; *Narayan et al., 2011*). So, it is also possible that the linear representation of behavior in *C. elegans* reflects linear circuitry in the brain.

We note that our exploration of non-linear models was not exhaustive. Although we tested a selection of non-linear models at the single neuron *Figure 3—figure supplement 5* and population level *Figure 3—figure supplement 4*, it is possible that a different non-linear model would perform better. And it is also possible that one of the non-linear models we did test would perform better with more training data. Complex models, including non-linear models, tend to have more parameters and are therefore prone to overfitting when trained on limited data. If a non-linear model performed poorly on our held-out data due to overfitting, it may perform better when trained with longer recordings. Poor performance here therefore does not inherently preclude a non-linear model from being useful for describing behavior signals in the *C. elegans* nervous system. Future work with longer recordings or the ability to aggregate training across multiple recordings is needed to better evaluate whether more complex models would outperform the simple linear decoder.

The types of signals used by the decoder are informative. The decoder uses a mix of neural activity signals and their temporal derivatives. This is consistent with prior reports that for some neurons, like AVA, it is the temporal derivative of activity that correlates with aspects of locomotion (*Kato et al., 2015*) while for other neurons, such as AIY, it is the activity itself (*Luo et al., 2014*). Temporal derivatives are one way for a model to incorporate temporal information. That the temporal derivative is informative, suggests that the nervous system cares not only about activity at this instant in time, but also about preceding moments. Future models could explicitly assign weights to

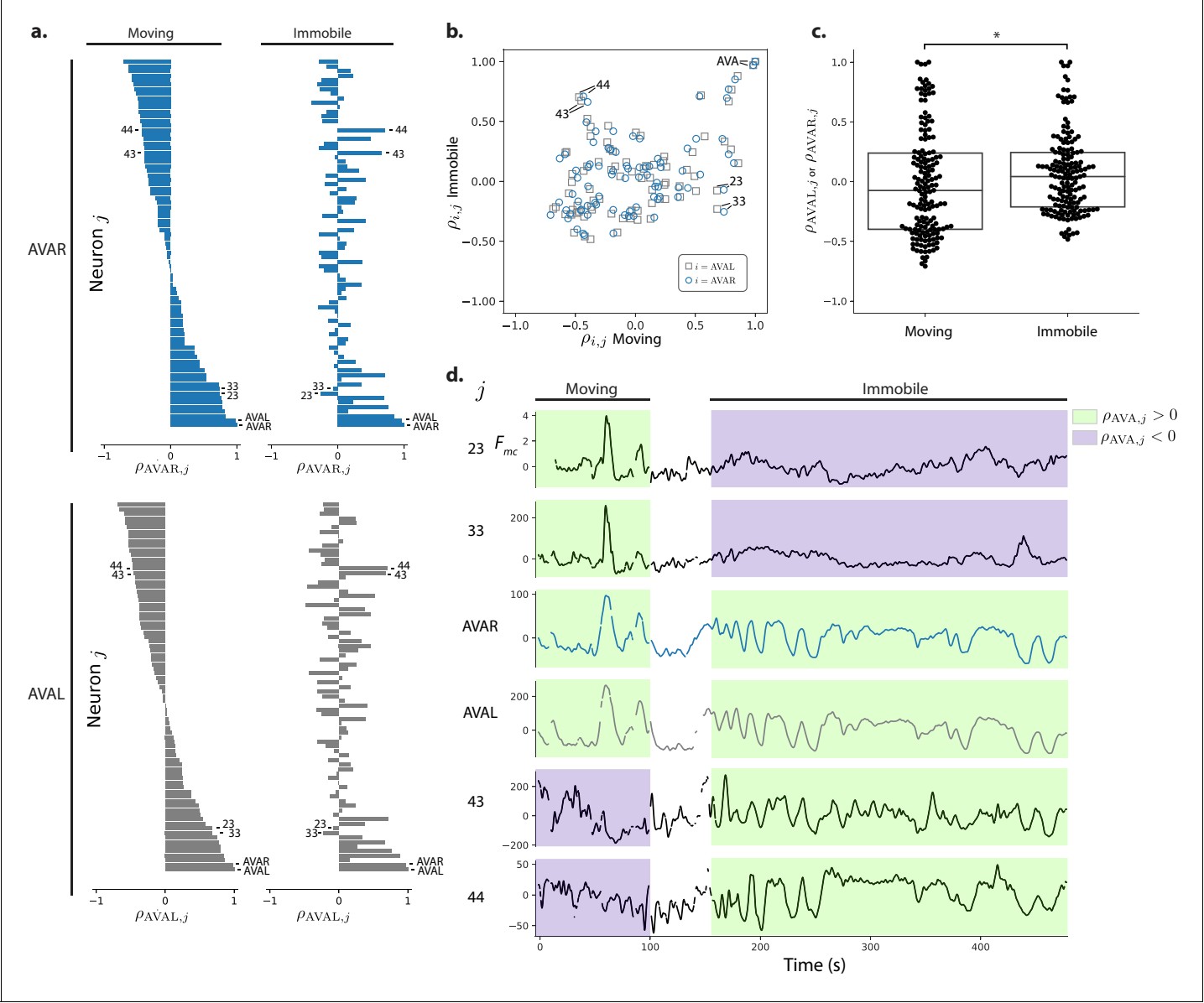

**Figure 8.** Correlations with respect to AVAL and AVAR during movement and immobilization. (**a**) The Pearson's correlation of each neuron's activity to AVAR and AVAL is shown during movement and immobilization. Selected neurons are numbered as in *Figure 7* (same recording, AML310_E). Neurons are sorted according to their correlation during movement. (**b**) Scatter plot shows relation between a neuron's correlation to AVA during movement and its correlation during immobilization. Gray squares and blue circles indicate correlation to AVAL and AVAR, respectively. (**c**) On average, neurons become more positively correlated to AVA upon immobilization, p = 0.019 Wilcoxon ranked test. Box shows median and interquartile range. (**d**) Activity traces of selected neurons are shown time aligned to AVA. Green and purple shading indicate positive or negative correlation to AVA, respectively.

the same neural activity at more time points, although this would likely require more training data to avoid overfitting.

Some of the signals used by the decoder were consistently correlated with locomotion throughout the duration of the recording. But other signals used by the decoder had pronounced peaks of activity that were relevant only for particular aspects. For example, some neurons had peaks that corresponded only to ventral but not dorsal turns, or vice versa. This is consistent with neurons such as RIVL/R that are active during ventral turns (*Wang et al., 2020*) or the SMDDs or SMDVs that have activity peaks during either dorsal or ventral head bends, respectively (*Hendricks et al., 2012*; *Shen et al., 2016*; *Kaplan et al., 2020*). Intriguingly, the decoder sometimes used signals that had peaks of activity only for particular instances of what appeared to be the same behavior motif, for

example one reversal event but not another. By summing up contributions from multiple neurons, the population model was able to capture relevant activity from different neurons at different times to decode all instances of the behavioral motif.

One possible explanation is that superficially similar behavioral features like turns may actually consist of different underlying behaviors. For example, seemingly similar turns, on closer inspection, can be further subdivided into distinct groups (*Broekmans et al., 2016*). The neural representation associated with a motif may also depend on its behavioral context, including the behaviors that follow or proceed it. For example, the temporal derivative of activity of AIB has been shown to be elevated during those reversals that are followed by turns compared to those followed by forward locomotion (*Wang et al., 2020*). The population may contain a variety of such neurons, each tuned to only a specific context of a given behavior, which would give rise to the neurons used by the decoder that are seemingly tuned to some instances of a motif and not others. The granularity with which to classify behaviors and how to take into account context and behavioral hierarchies remains an active area of research in *C. elegans* (*Liu et al., 2018*; *Kaplan et al., 2020*) and in other model systems (*Berman et al., 2016*; *Datta et al., 2019*). Ultimately, finding distinct neural signals may help inform our understanding of distinct behavior states and vice versa.

A related possibility is that the same behavior motifs are initiated in the head through different neural pathways. Previous work has suggested that activity in either of two different sets of head interneurons, AVA/AVE/AVD or AIB/RIM, are capable of inducing reversal behavior independently (*Piggott et al., 2011*). If these or other neurons were active only for a subset of reversals, it could explain why some neurons seem to have activity relevant for some behavioral instances but not others. The specific neurons listed in *Piggott et al., 2011* do not fit the pattern observed in our measurements in part because we observe that AVA shows expected activity transients for nearly all large reversals. But it is possible that other neurons in the two subsets, or indeed other subsets of neurons, provide relevant activity for only some instances of a behavior.

Similarly, different sensory modalities such as mechanosensation (*Chalfie et al., 1985*), thermosensation (*Croll, 1975*) and chemosensation (*Ward, 1973*) are known to evoke common behavioral outputs via sensory pathways that have both common and distinct elements. For example, both polymodal nociceptive stimuli detected from ASH (*Mellem et al., 2002*) and anterior mechanosensory stimuli detected from soft touch neurons ALM and AVM (*Wicks and Rankin, 1995*) activate reversals through shared circuitry containing AVA, among other common neurons. It is possible that the neural activities we observe for different behavioral motifs reflect sensory signals that arrive through different sensory pathways to evoke a common downstream motor response.

By inspecting the neural weights assigned by our model, we found that only a fraction of neurons are necessary for the model to achieve 90% of its performance. Sub-populations of neurons with modest overlap contribute the majority of information for decoding velocity and curvature, respectively. This is consistent with other reports, including recent work suggesting that turning and reverse circuits are largely distinct modules except for a select few neurons, such as RIB, which may be involved in both (*Wang et al., 2020*). Future studies using newly developed methods for identifying neurons (*Yemini et al., 2021*) are needed to reveal the identities of those neurons weighted by the decoder for decoding velocity, curvature, or both.

That *C. elegans* neural dynamics exhibit different correlation structure during movement than during immobilization has implications for neural representations of locomotion. For example, it is now common to use dimensionality reduction techniques like PCA to search for low-dimensional trajectories or manifolds that relate to behavior or decision making in animals undergoing movement (*Churchland et al., 2012*; *Harvey et al., 2012*; *Shenoy et al., 2013*) or in immobilized animals undergoing fictive locomotion (*Briggman et al., 2005*; *Kato et al., 2015*). PCA critically depends on the correlation structure to define its principal components. In *C. elegans*, the low-dimensional neural trajectories observed in immobilized animals undergoing fictive locomotion, and the underlying correlation structure that defines those trajectories, are being used to draw conclusions about neural dynamics of actual locomotion. Our measurements suggest that to obtain a more complete picture of *C. elegans* neural dynamics related to locomotion, it will be helpful to probe neural state space trajectories recorded during actual locomotion: both because the neural dynamics themselves may differ during immobilization, but also because the correlation structure observed in the network, and consequently the relevant principal components, change upon immobilization. These changes may

be due to proprioception (*Wen et al., 2012*), or due to different internal states associated with fictive versus actual locomotion.

# Materials and methods

## Key resources table

| Reagent type (species) or resource | Designation | Source or reference | Identifiers | Additional information |
|---|---|---|---|---|
| Strain, strain background (*C. elegans*) | AML310 | this work | | Details in *Table 2* |
| Strain, strain background (*C. elegans*) | AML32 | *Nguyen et al., 2017* | RRID:WBI-STRAIN:WBStrain00000192 | |
| Strain, strain background (*C. elegans*) | AML18 | *Nguyen et al., 2016* | RRID:WBI-STRAIN:WBStrain00000191 | |

### Strains

Three strains were used in this study, see *Table 2*. AML32 (*Nguyen et al., 2017*) and AML310 were used for calcium imaging. AML18 (*Nguyen et al., 2016*) served as a calcium insensitive control. Strain AML310 is similar to AML32 but includes additional labels to identify AVA neurons. AML310 was generated by injecting 30 ng/ul of P*rig-3*::tagBFP plasmid into AML32 strains (wtfls5[P*rab-3*:: NLS ::GCaMP6s; P*rab-3*::NLS::tagRFP]). AML310 worms were selected and maintained by picking individuals expressing BFP fluorescence in the head. Animals were cultivated in the dark on NGM plates with a bacterial lawn of OP50.

### Whole brain imaging

Whole brain imaging in moving animals

Whole brain imaging of moving animals was performed as described previously (*Nguyen et al., 2016*). *Table 3* lists all recording used in the study, and *Table 4* cross-lists the recordings according to figure. Briefly, adult animals were placed on an imaging plate (a modified NGM media lacking cholesterol and with agarose in place of agar) and covered with mineral oil to provide optical index matching to improve contrast for behavior imaging (*Leifer et al., 2011*). A coverslip was placed on top of the plate with 100 m plastic spacers between the coverglass and plate surface. The coverslip was fixed to the agarose plate with valap. Animals were recorded on a custom whole brain imaging system, which simultaneously records four video streams to image the calcium activity of the brain while simultaneously capturing the animal's behavior as the animal crawls on agar in two-dimensions. We record ×10 magnification darkfield images of the body posture, ×10x magnification fluorescence images of the head for real-time tracking, and two ×40 magnification image streams of the neurons in the head, one showing tagRFP and one showing either GCaMP6s, GFP, or BFP. The ×10 images are recorded at 50 frames/s, and the 40x fluorescence images are recorded at a rate of 200 optical slices/s, with a resulting acquisition rate of 6 head volumes/s. Recordings were stopped when the animal ran to the edge of the plate, when they left the field of view, or when photobleaching decreased the contrast between tag-RFP and background below a minimum level. Intensity of excitation light for fluorescent imaging was adjusted from recording to recording to achieve different tradeoffs between fluorescence intensity and recording duration.

**Table 2.** Strains used.
Associated Research Resource Identifiers are listed in Key Resources.

| Strain | Genotype | Expression | Role | Reference |
|---|---|---|---|---|
| AML310 | wtfls5[P*rab-3*::NLS::GCaMP6s; P*rab-3*:: NLS::tagRFP]; wtfEx258 [P*rig-3*::tagBFP:: unc-54] | tag-RFP and GCaMP6s in neuronal nuclei; BFP in cytoplasm of AVA and some pharyngeal neurons (likely I1, I4, M4 and NSM) | Calcium imaging with AVA label | This Study |
| AML32 | wtfls5[P*rab-3*::NLS::GCaMP6s; P*rab-3*:: NLS::tagRFP] | tag-RFP and GCaMP6s in neuronal nuclei | Calcium imaging | *Nguyen et al., 2017* |
| AML18 | wtfls3[P*rab-3*::NLS::GFP, P*rab-3*::NLS:: tagRFP] | tag-RFP and GFP in neuronal nuclei | Control | *Nguyen et al., 2016* |

**Table 3.** Recordings used in this study.

| Unique ID | Strain | Duration (mins) | Notes |
|---|---|---|---|
| AML310_A | AML310 | 4 | Ca$^{2+}$ imaging w/ AVA label, moving |
| AML310_B | | 4 | |
| AML310_C | | 4 | |
| AML310_D | | 4 | |
| AML310_E | AML310 | 8 | Ca$^{2+}$ imaging w/ AVA label, moving-to-immobile |
| AML310_F | | 8 | |
| AML310_G | AML310 | 15 | Ca$^{2+}$ imaging w/ AVA label, immobile |
| AML32_A | AML32 | 11 | Ca$^{2+}$ imaging, moving |
| AML32_B | | 11 | |
| AML32_C | | 10 | |
| AML32_D | | 11 | |
| AML32_E | | 4 | |
| AML32_F | | 5 | |
| AML32_G | | 4 | |
| AML32_H | AML32 | 13 | Ca$^{2+}$ imaging, moving-to-immobile |
| AML18_A | AML18 | 10 | GFP control, moving |
| AML18_B | | 10 | |
| AML18_C | | 7 | |
| AML18_D | | 5 | |
| AML18_E | | 5 | |
| AML18_F | | 6 | |
| AML18_G | | 9 | |
| AML18_H | | 6 | |
| AML18_I | | 7 | |
| AML18_J | | 6 | |
| AML18_K | | 6 | |

Moving recordings had to meet the following criteria. The animal had to be active and the recording had to be at least 200 s. The tag-RFP neurons also had to be successfully segmented and tracked via our analysis pipeline.

## Moving to immobile transition experiments

Adult animals were placed in a PDMS microfluidic artificial dirt style chip (*Lockery et al., 2008*) filled with M9 medium where the animal could crawl. The chip was imaged on the whole brain imaging system. A computer controlled microfluidic pump system delivered either M9 buffer or M9 buffer with the paralytic levamisole or tetramisole to the microfluidic chip. Calcium activity was recorded from the worm as M9 buffer flowed through the chip with a flow rate of order a milliliter a minute. Partway through the recording, the drug buffer mixture was delivered at the same flow rate. At the conclusion of the experiment for AML310 worms, BFP was imaged.

Different drug concentrations were tried for different recordings to find a good balance between rapidly immobilizing the animal without also inducing the animal to contract and deform. Paralytic concentrations used were: 400 µM for AML310_E, 100 µM for AML310_F, and 5 µM for AML32_H.

Recordings were performed until a recording achieved the following criteria for inclusion: (1) the animal showed robust locomotion during the moving portion of the recording, including multiple reversals. (2) The animal quickly immobilized upon application of the drug. (3) The animal remained immobilized for the remainder of the recording except for occasional twitches, (4) the immobilization portion of the recording was of sufficient duration to allow us to see multiple cycles of the

**Table 4.** List of recordings included in each figure.

| Figure | Recordings |
|---|---|
| *Figure 1*; *Figure 1—figure supplement 1*; *Figure 1—figure supplement 2*; | AML310_A |
| *Figure 1—figure supplement 3* | AML310_A-D, AML32_A-G, AML18_A-K |
| *Figure 2a,b*; *Figure 2—figure supplement 1* | AML310_A |
| *Figure 2c* | AML310_A-D |
| *Figure 3a–d* | AML310_A |
| *Figure 3e,f* | AML310_A-D, AML32_A-G, AML18_A-K |
| *Figure 3—figure supplement 1*; *Figure 3—figure supplement 2*; *Figure 3—figure supplement 4*; *Figure 3—figure supplement 5* | AML310_A-D, AML32_A-G |
| *Figure 3—figure supplement 3* | AML18_A-K |
| *Figure 4* | AML32_A |
| *Figure 5*; *Figure 5—figure supplement 1*; *Figure 5—figure supplement 2* | AML310_A |
| *Figure 5—figure supplement 3* | AML32_A |
| *Figure 6a,b*; *Figure 6—video 1* | AML310_A |
| *Figure 6c* | AML310_A-D, AML32_A-G |
| *Figure 7a–f* | AML310_E |
| *Figure 7g* | AML310_A-F, AML32_A-H |
| *Figure 7—figure supplement 1* | AML32_H |
| *Figure 7—figure supplement 2* | AML310_G |
| *Figure 8* | AML310_E |

stereotyped neural state space trajectories if present and (5) for strain AML310, neurons AVAL and AVAR were required to be visible and tracked throughout the entirety of the recording. For the statistics of correlation structure in *Figure 7f*, recording AML310_F was also included even though it did not meet all criteria (it lacked obvious reversals).

## Whole brain imaging in immobile animals

We performed whole brain imaging in adult animals immobilized with 100 nm polystyrene beads (*Kim et al., 2013*). The worms were then covered with a glass slide, sealed with valap, and imaged using the Whole Brain Imager.

## Neuron segmentation, tracking, and fluorescence extraction

Neurons were segmented and tracked using the Neuron Registration Vector Encoding (NeRVE) and clustering approach described previously (*Nguyen et al., 2017*) with minor modifications which are highlighted below. As before, video streams were spatially aligned with beads and then synchronized using light flashes. The animals' posture was extracted using an active contour fit to the ×10 magnification darkfield images. But in a departure from the method in *Nguyen et al., 2017*, the high magnification fluorescent images are now straightened using a different centerline extracted directly from the fluorescent images. As in *Nguyen et al., 2017*, the neural dynamics were then extracted by segmenting the neuronal nuclei in the red channel and straightening the image according to the body posture. Using repeated clustering, neurons are assigned identities over time. The GCaMP signal was extracted using the neural positions found from tracking. The pipeline returns datasets containing RFP and GCaMP6s fluorescence values for each successfully tracked neuron over time, and the centerline coordinates describing the posture of the animal over time. These are subsequently processed to extract neural activity or behavior features.

The paralytic used in moving-to-immobile recordings (*Figure 7*) caused the animal's head to contract, which would occasionally confuse our tracking algorithm. In those instances the automated NeRVE tracking and clustering was run separately on the moving and immobile portions of the

recording (before and after contraction), and then a human manually tracked neurons during the transition period (1–2 min) so as to stitch the moving and immobile tracks together.

## Photobleaching correction, outlier detection, and pre-processing

The raw extracted RFP or GCaMP fluorescent intensity timeseries were preprocessed to correct for photobleaching. Each time-series was fit to a decaying exponential. Those that were well fit by the exponential were normalized by the exponential and then rescaled to preserve the timeseries' original mean and variance as in *Chen et al., 2019*. Timeseries that were poorly fit by an exponential were left as is. If the majority of neurons in a recording were poorly fit by an exponential, this indicated that the animal may have photobleached prior to the recording and the recording was discarded.

Outlier detection was performed to remove transient artifacts from the fluorescent time series. Fluorescent time points were flagged as outliers and omitted if they met any of the following conditions: the fluorescence deviated from the mean by a certain number of standard deviations ($F < -2\sigma$ or $F > 5\sigma$ for RFP; $|F| > 5\sigma$ for GCaMP); the RFP fluorescence dropped below a threshold; the ratio of GCaMP to RFP fluorescence dropped below a threshold; a fluorescence timepoint was both preceded by and succeeded by missing timepoints or values deemed to be outliers; or if the majority of other neurons measured during the same volume were also deemed to be outliers.

Fluorescent time series were smoothed by convolution with a Gaussian ($\sigma = 0.83$ s) after interpolation. Omitted time points, or gaps where the neuron was not tracked, were excluded from single-neuron analyses, such as the calculation of each neuron's tuning curve. It was not practical to exclude missing time points from the population-level analyses such as linear decoding. In these population-level analyses, interpolated values were used. Time points in which the majority of neurons had missing fluorescent values were excluded, even in population level analyses. Those instances are shown as white vertical stripes in the fluorescent activity heatmaps, for example, as visible in *Figure 1*.

## Motion-correction

We used the GCaMP fluorescence together with the RFP fluorescence to calculate a motion corrected fluorescence, $F_{mc}$ used through the paper. Note sometimes the subscript $_{mc}$ is omitted for brevity. Motion and deformation in the animal's head introduce artifacts into the fluorescent timeseries. We assume that these artifacts are common to both GCaMP and RFP fluorescence, up to a scale factor, because both experience the same motion. For example, if a neuron is compressed during a head bend, the density of both GCaMP and RFP should increase, causing an increase in the fluorescence in both time-series. We expect that the RFP time series is entirely dominated by artifacts because, in the absence of motion, the RFP fluorescent intensity would be constant. If we further assume that motion artifacts are additive, then a simple correction follows naturally. To correct for motion in the GCaMP fluorescence $G$, we subtract off a scaled RFP fluorescence, $R$,

$$F_{\mathrm{mc}} = (G - \alpha R) - \langle G - \alpha R \rangle, \tag{2}$$

where $\alpha$ is a scaling factor that is fit for each neuron so as to minimize $\sum (G(t) - \alpha R(t))^2$. This approach has similarities to *Tai et al., 2004*. The final motion corrected signal $F_{\mathrm{mc}}$ is mean-subtracted.

When presenting heatmaps of calcium activity, we use the colormap to convey information about the relative presence of calcium activity compared to motion artifact in the underlying recording. The limits on the colormap are determined by the uncorrected green fluorescent timeseries, specifically the 99th percentile of $\pm|G - \langle G \rangle|$ of all neurons at all time points in the recording. With this colormap, recordings in which the neurons contain little signal compared to motion artifact will appear dim, while recordings in which neurons contain signal with large dynamics compared to the motion artifact will appear bright.

## Temporal derivative

The temporal derivatives of motion corrected neuron signals are estimated using a Gaussian derivative kernel of width 2.3 s. For brevity we denote this kernel-based estimate as $\frac{dF}{dt}$.

## Identifying AVA

AVAL and AVAR were identified in recordings of AML310 by their known location and the presence of a BFP fluorescent label expressed under the control of the *rig-3* promoter. BFP was imaged immediately after calcium imaging was completed, usually while the worm was still moving. To image BFP, a 488 nm laser was blocked and the worm was then illuminated with 405 nm laser light. In one of the recordings, only one of the two AVA neurons was clearly identifiable throughout the duration of the recording. For that recording, only one of the AVA neurons was included in analysis.

## Measuring and representing locomotion

To measure the animal's velocity $v$, we first find the velocity vector that describes the motion of a point on the animal's centerline 15% of its body length back from the tip of its head. We then project this velocity vector onto a head direction vector of unit length. The head direction is taken to be the direction between two points along the animal's centerline, 10% and 20% posterior of the tip of the head.

To calculate this velocity, the centerline and stage position measurements were first Hampel filtered and then interpolated onto a common time axis of 200 Hz (the rate at which we query stage position). Velocity was then obtained by convolving the position with the derivative of a Gaussian with $\sigma = 0.5$ s.

To measure the animal's average curvature $\langle \kappa \rangle$ at each time point, we calculated the curvature $d\theta/ds$ at each of 100 segments along the worm's centerline, where $s$ refers to the arc length of the centerline. We then took the mean of the curvatures of the middle segments that span an anterior-posterior region from 15% to 80% along the animal's centerline. This region was chosen to exclude curvature from small nose deflections (sometimes referred to as foraging) and to exclude the curvature of the tip of the tail.

## Relating neural activity to behavior

### Tuning curves

The Pearson's correlation coefficient $\rho$ is reported for each neurons' tuning, as in *Figure 1d,e*. To reject the null hypothesis that a neuron is correlated with behavior by chance we took a shuffling approach and applied a Bonferroni correction for multiple hypothesis testing. We shuffled our data in such a way as to preserve the correlation structure in our recording. To calculate the shuffle, each neuron's activity was time-reversed and circularly shifted relative to behavior by a random time lag and then the Pearson's correlation coefficient was computed. Shuffling was repeated for each neuron in a recording $M$ times to build up a distribution of $M \times N$ values of $\rho$, where $N$ is the number of neurons in the recording. For AML310_A, we shuffled each neuron in the recording $M = 5000$ times. For other datasets we shuffled each neuron $M = 500$ times. To reject the null hypothesis at 0.05% confidence, we apply a Bonferonni correction such that a correlation coefficient greater than $\rho$ (or less than, depending on the sign) must have been observed in the shuffled distribution with a probability less than $0.05/(2N)$. The factor of $2N$ arises from accounting for multiple hypothesis testing for tuning of both $F$ and $dF/dt$ for each neuron.

### Population model

We use a ridge regression (*Hoerl and Kennard, 1970*) model to decode behavior signals $y(t)$ (the velocity and the body curvature). The model prediction is given by a linear combination of neural activities and their time derivatives,

$$\hat{y}(t) = \sum_i \left( W_{F,i} F_i(t) + W_{\frac{dF}{dt},i} \frac{dF_i}{dt}(t) \right) + \beta. \tag{3}$$

Note here we are omitting the $\sim_{\mathrm{mc}}$ subscript for convenience, but these still refer to the motion corrected fluorescence signal.

We scale all these features to have zero mean and unit variance, so that the magnitudes of weights can be compared to each other. To determine the parameters $\{W_{F,i}, W_{\frac{dF}{dt},i}, \beta\}$, we hold out a test set comprising the middle 40% of the recording, and use the remainder of the data for training. We minimize the cost function

$$C = \sum_{t \in \text{Train}} (y(t) - \hat{y}(t))^2 + \lambda \sum_{i} \left( W_{F,i}^2 + W_{\frac{dF}{dt},i}^2 \right). \tag{4}$$

The hyperparameter $\lambda$ sets the strength of the ridge penalty in the second term. We choose $\lambda$ by splitting the training set further into a second training set and a cross-validation set, and training on the second training set with various values of $\lambda$. We choose the value which gives the best performance on the cross-validation set.

To evaluate the performance of our model, we use a mean-subtracted coefficient of determination metric, $R_{\text{MS}}^2$, on the test set. This is defined by

$$R_{\text{MS}}^2(y, \hat{y}) = R^2(y - \langle y \rangle, \hat{y} - \langle \hat{y} \rangle), \tag{5}$$

where we use the conventional definition of $R^2$, defined here for an arbitrary true signal $z$ and corresponding model prediction $\hat{z}$:

$$R^2(z, \hat{z}) = 1 - \frac{\sum_{t \in \text{Test}} (z(t) - \hat{z}(t))^2}{\sum_{t \in \text{Test}} (z(t) - \langle z(t) \rangle)^2}. \tag{6}$$

Note that $R_{\text{MS}}^2$ can take any value on $(-\infty, 1]$.

## Restricted models

To assess the distribution of locomotive information throughout the animal's brain, we compare with two types of restricted models. First, we use a Best Single Neuron model in which all but one of the coefficients $\{W_{F,i}, W_{\frac{dF}{dt},i}\}$ in *Equation (3)* are constrained to vanish. We thus attempt to represent behavior as a linear function of a single neural activity, or its time derivative These models are shown in *Figure 3*. Second, after training the population model, we sort the neurons in descending order of $\max(|W_{F,i}|, |W_{\frac{dF}{dt},i}|)$. We then construct models using a subset of the most highly weighted neurons, with the relative weights on their activities and time derivatives fixed by those used in the population model. The performance of these truncated models can be tabulated as a function of the number of neurons included to first achieve a given performance, as shown in *Figure 6*. Note that when reporting fraction of total model performance for this partial model, we evaluate performance on the entire dataset (held-out and training, denoted $R_{\text{MS,all}}^2$) because all relative weights for the model have already been frozen in place and there is no risk of overfitting.

## Alternative models

The population model used throughout this work refers to a linear model with derivatives using ridge regression. In *Figure 3—figure supplement 4*, we show the performance of seven alternative population models at decoding velocity for our exemplar recording. The models are summarized in *Table 5*. Many of these models perform similarly to the linear population model used throughout the paper. Our chosen model was selected both for its relative simplicity and because it showed one of the highest mean performances at decoding velocity across recordings.

*Figure 3—figure supplement 4a–b* show the model we use throughout the paper, and the same model but with only fluorescence signals (and not their time derivatives) as features. The latter model attains a slightly lower score of $R_{\text{MS}}^2 = 0.60$. Note that while adding features is guaranteed to improve performance on the training set, performance on the held-out test set did not necessarily have to improve. Nonetheless, we generally found that including the time derivatives led to better predictions on the test set.

*Figure 3—figure supplement 4c–d* show a variant of the linear model where we add an acceleration penalty to the model error. Our cost function becomes (*Equation 4*).

$$C = \sum_{t \in \text{Train}} \left( (y(t) - \hat{y}(t))^2 + \mu \left( \frac{dy}{dt}(t) - \frac{d\hat{y}}{dt}(t) \right)^2 \right) + \lambda \sum_{i} \left( W_{F,i}^2 + W_{\frac{dF}{dt},i}^2 \right), \tag{7}$$

where the derivatives $\frac{dy}{dt}$ and $\frac{d\hat{y}}{dt}$ are estimated using a Gaussian derivative filter. The parameter $\mu$ is

**Table 5.** Alternative models explored.

Most are linear models, using either the Ridge or ElasticNet regularization. In some cases, we add an additional term to the cost function which penalizes errors in the temporal derivative of model output (which, for velocity models, corresponds to the error in the predicted acceleration). For features, we use either the neural activities alone, or the neural activities together with their temporal derivatives. We also explore two nonlinear models: MARS *Friedman, 1991*, and a shallow decision tree which chooses between two linear models.

| Model | Penalty | Features | Number of parameters |
|---|---|---|---|
| Linear | Ridge | $F$ and $dF/dt$ | $2N_n + 1$ |
| Linear | Ridge | $F$ | $N_n + 1$ |
| Linear | Ridge + Acceleration Penalty | $F$ and $dF/dt$ | $2N_n + 1$ |
| Linear | Ridge + Acceleration Penalty | $F$ | $N_n + 1$ |
| Linear | ElasticNet | $F$ and $dF/dt$ | $2N_n + 1$ |
| Linear | ElasticNet | $F$ | $N_n + 1$ |
| MARS (nonlinear) | MARS | $F$ and $dF/dt$ | variable |
| Linear with Decision Tree (nonlinear) | Ridge | $F$ and $dF/dt$ | $4N_n + 9$ |

set to 10. For our exemplar recording, adding the acceleration penalty hurts the model when derivatives are not included as features, but has little effect when they are.

*Figure 3—figure supplement 4e–f* show a variant where we use an ElasticNet penalty instead of a ridge penalty (*Zou and Hastie, 2005*). If we write the ridge penalty as the $L_2$ norm of the weight vector, so that

$$\lambda \sum_i \left( W_{F,i}^2 + W_{\frac{dF}{dt},i}^2 \right) \equiv \lambda W_2^2, \tag{8}$$

the ElasticNet penalty is defined by

$$\lambda \left( r W_1 + (1-r) W_2^2 \right), \tag{9}$$

where

$$W_1 = \sum_i \left( \left| W_{F,i} \right| + \left| W_{\frac{dF}{dt},i} \right| \right) \tag{10}$$

is the $L_1$ norm of the weight vector. The quantity $r$ is known as the $L_1$ ratio, and in *Figure 3—figure supplement 4* it is set to $10^{-2}$. We have also tried setting $r$ via cross-validation, and found similar results.

*Figure 3—figure supplement 4g* uses the multivariate adaptive regression splines (MARS) model (*Friedman, 1991*). The MARS model incorporates nonlinearity by using rectified linear functions of the features, or products of such functions. Generally, they have the advantage of being more flexible than linear models while remaining more interpretable than a neural network or other more complicated nonlinear model. However, we find that MARS somewhat underperforms a linear model on our data.

*Figure 3—figure supplement 4h* uses a decision tree classifier trained to separate the data into forward-moving and backward-moving components, and then trains separate linear models on each component. For our exemplar recording, this model performs slightly better than the model we use throughout the paper. This is likely a result of the clear AVAR signal in *Figure 2*, which can be used by the classifier to find the backward-moving portions of the data. Across all our recordings, this model underperforms the simple linear model.

## Correlation structure analysis

The correlation structure of neural activity was visualised as the correlation matrix, $\rho_{i,j}$. To observe changes in correlation structure, a correlation matrix for the moving portion of the recording was calculated separately from the immobile portion. The time immediately following delivery of the

paralytic when the animal was not yet paralzed was excluded (usually one to two minutes). To quantify the magnitude of the change in correlation structure, a dissimilarity metric was defined as the root mean-squared change in each neuron's pairwise correlations, $\sqrt{\left\langle (\rho'_{i,j} - \rho_{i,j})^2 \right\rangle}$. As a control, changes to correlation structure were measured in moving animals. In this case the correlation structure of the first 30% of the recording was compared to the correlation structure of latter 60% of the recording, so as to mimic the relative timing in the moving-to-immobile recordings.

## Software

Analysis scripts are available at https://github.com/leiferlab/PredictionCode (*Leifer, 2021*, copy archived at swh:1:rev:ca59416112a9c10a8d6a3179092a7d3c888bcd4e).

## Data

Data from all experiments including calcium activity traces and animal pose and position are publicly available at https://doi.org/10.17605/OSF.IO/DPR3H.

## Acknowledgements

Thanks to Sandeep Kumar and Kevin Chen for critical comments on the manuscript. This work was supported in part by the National Science Foundation, through the Center for the Physics of Biological Function (PHY-1734030 to JWS, AML, KMH) and an NSF CAREER Award (IOS-1845137 to AML) and by the Simons Foundation (SCGB #324285, and SCGB #543003, AML). ANL is supported by a National Institutes of Health institutional training grant NIH T32 MH065214 through the Princeton Neuroscience Institute. FR was supported by the Swartz Foundation via the Swartz Fellowship for Theoretical Neuroscience. Strains are distributed by the CGC, which is funded by the NIH Office of Research Infrastructure Programs (P40 OD010440).

## Additional information

### Funding

| Funder | Grant reference number | Author |
| --- | --- | --- |
| National Science Foundation | IOS-1845137 | Andrew M Leifer |
| National Science Foundation | PHY-1734030 | Kelsey M Hallinen<br>Joshua W Shaevitz<br>Andrew M Leifer |
| National Institutes of Health | MH065214 | Ashley Linder |
| Simons Foundation | 324285 | Andrew M Leifer |
| Swartz Foundation | | Francesco Randi |

The funders had no role in study design, data collection and interpretation, or the decision to submit the work for publication.

### Author contributions

Kelsey M Hallinen, Data curation, Formal analysis, Investigation, Visualization, Methodology, Writing - original draft, Writing - review and editing; Ross Dempsey, Monika Scholz, Software, Formal analysis, Investigation, Visualization, Methodology, Writing - original draft, Writing - review and editing; Xinwei Yu, Ashley Linder, Resources, Software, Formal analysis, Investigation, Methodology, Writing - review and editing; Francesco Randi, Resources, Methodology, Writing - review and editing, Developed instrumentation; Anuj K Sharma, Resources, Methodology, Writing - review and editing, Generated all transgenics; Joshua W Shaevitz, Conceptualization, Supervision, Funding acquisition, Writing - review and editing; Andrew M Leifer, Conceptualization, Software, Formal analysis, Supervision, Funding acquisition, Visualization, Writing - original draft, Project administration, Writing - review and editing

## Author ORCIDs
Kelsey M Hallinen ⓘD https://orcid.org/0000-0003-4081-6699
Ross Dempsey ⓘD https://orcid.org/0000-0002-0881-8814
Monika Scholz ⓘD https://orcid.org/0000-0003-2186-410X
Xinwei Yu ⓘD https://orcid.org/0000-0002-8699-3546
Francesco Randi ⓘD https://orcid.org/0000-0002-6200-7254
Anuj K Sharma ⓘD https://orcid.org/0000-0001-5061-9731
Joshua W Shaevitz ⓘD http://orcid.org/0000-0001-8809-4723
Andrew M Leifer ⓘD https://orcid.org/0000-0002-5362-5093

## Decision letter and Author response
Decision letter https://doi.org/10.7554/eLife.66135.sa1
Author response https://doi.org/10.7554/eLife.66135.sa2

# Additional files

## Supplementary files
• Transparent reporting form

## Data availability
Data associated with this manuscript has been deposited in a publicly accessible repository hosted by the Open Science Framework at https://doi.org/10.17605/OSF.IO/DPR3H.

The following dataset was generated:

| Author(s) | Year | Dataset title | Dataset URL | Database and Identifier |
|---|---|---|---|---|
| Hallinen KM, Dempsey R, Scholz M, Yu X, Linder A, Randi F, Sharma A, Shaevitz JW, Leifer AM | 2020 | Decoding locomotion from population neural activity in moving C. elegans | https://doi.org/10.17605/OSF.IO/DPR3H | Open Science Framework, 10.17605/OSF.IO/R5TB3 |

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
