## [Decision Letter]

**Acceptance summary:**

This paper will be of interest to a wide range of systems neuroscientists seeking to understanding the relationship between neuronal activity and behavior. Building on previous technical advances in brain-wide imaging of neuronal activity (Ca signals) in freely moving animals (*Caenorhabditis elegans*), it demonstrates that a linear regression model is sufficient reconstruct key parameters of locomotion – velocity and body curvature – from the imaging data and documents differences in activity between freely moving and immobilized worms.

**Decision letter after peer review:**

Thank you for submitting your article "Decoding locomotion from population neural activity in moving *C. elegans*" for consideration by *eLife*. Your article has been reviewed by 3 peer reviewers, and the evaluation has been overseen by a Reviewing Editor and Ronald Calabrese as the Senior Editor. The reviewers have opted to remain anonymous.

Essential Revisions:

1) Please address concerns by Reviewers #1 and #2 about identifying eigenworms with velocity and curvature (detailed in Recommendations for the authors).

2) Please address the questions with respect to tuning and noise that are raised by Reviewer #2 (detailed in Recommendations for the authors).

3) Both Reviewer #1 and #3 (detailed in Recommendations for the authors) require that you address your conclusion that the population decoder outperforms the best single neuron. Is this a meaningful comparison, and how should such coding be interpreted?

4) The concerns of Reviewers #2 and #3 about the significance of the distribution of weights assigned by the decoder for how behavior is represented in the brain should be addressed (detailed in Recommendations for the authors).

5) All Reviewers (detailed in Recommendations for the authors) have strong suggestions for reorganizing the text and amplifying and deepening Introduction and Discussion. Reviewer #3's concerns about the functional implications of the decoding should be addressed. Limitations of the analysis should be clearly addressed in Discussion.

*Reviewer #1 (Recommendations for the authors):*

I hope that the authors focus on improving results and discussions sections of their strength (see above), including additional analyses, precise terminology, simplified statements, clarified discussions, and perhaps structural reorganization. I have a few concerns that I ask them to address or respond to, so that this work can be appreciated by and benefit the field. They are raised below, and should be viewed as suggestions for this purpose.

(1) Line 71-85: This first Results section (which lacks a title) is a brief definition of the locomotion features for velocity and curvature as used throughout the paper.

I am uncomfortable with the brevity of the introduction and justification of using eigenworms to represent velocity and curvature. These are two widely used biological terms, and the introduction would confuse many readers and even misled them (in the case of 'curvature').

I share the authors' opinion on the deficiency of defining velocity by the animal's centroid displacement. However, they should be equally clear that their presentation for 'velocity' did not directly address this deficit: their analysis did not calculate and present the wave velocity – the speed of bending wave propagation – which would have the units of mm/sec or body lengths/sec as opposed to radians/sec.

Moreover, in Figure 1-Figure S1, the authors demonstrated that their eigenvalue-derived velocity was well correlated with that of centroid-derived velocity values. This, to me, was a good validation to justify their choice of parameters as a proxy for velocity in later analyses. However, the authors did not cite this validation figure as its purpose, but instead in the context of a statement for the weakness of the centroid-based velocity measure. This is a misleading manipulation of citation of the authors' results.

I have a bigger concern for referencing the third eigenworm as the 'curvature', specifically Lines 82-84 ("Here we report body curvature as a dimensionless quantity that captures bending in the dorsoventral plane, calculated by projecting the animal's body posture onto the third principal component of the eigenvalue decomposition."). To my understanding, this component best represents the body postures during turning. Their relationship with 'curvature' – which most would interpret not as a dimensionless quantity but as a precise measure of the degree of body bending per unit length – should be demonstrated similar to how the authors did so for velocity in Figure 1, Supplementary Figure 1. I personally consider it inappropriate to use 'curvature' when referring to the projections of the third eigenworm.

2) I found their motion correction important, interesting, and potentially useful to the community. The authors should definitely highlight it and elaborate in the text as a separate section instead of putting it away in Methods and at the end of the following Results section (Line 125: Population decoder outperforms best single neuron – this long result section can definitely benefit from 'de-mixing'.)

To me, it would be very helpful to show the example data for the authors' methods for motion correction, including the raw traces of GCaMP and RFP before and after they performed correction by their ICA analyses (e.g. I think that it did not work as well for AVAL in Figure 2b; knowing what the trace was like before the correction would help me to examine why). I also would be curious to know why these authors limited their ICA to give two components instead of collecting all components and subtracting the ones correlated with RFP. It would be good if authors treated the number of ICA components as a parameter and explored the choice of this parameter on the performance of motion correction. A discussion on systematic ways to estimate this parameter would also be very welcome.

3) Section 'Population decoder outperforms best single neuron' and Figure 3a.

Here I have trouble appreciating the significance of this comparison. Previous studies have shown that forward, backward, and turning are three separate motor motifs of *C. elegans* locomotion. It is possible that multiple neurons may participate in multiple motor behaviors, but it would be truly astonishing (to me at least) if a single neuron plays a dominating role of all motifs of locomotion. Given the state of the field, scientifically it would be much more meaningful to compare the performance of a population decoder to the combination of the four best single neurons e.g. the best for positive velocity, the best for negative velocity, the best for dorsal turning, and the best for ventral turning, instead of one single best neuron.

The authors could also make it clear to readers that due to the lack of knowledge of neuronal identity, as well as the fact that each recording was capturing ~2/3 of the total neuronal population, the best single neuron decoder in each recording was only 'relative' to the captured neuronal population, and likely differed per recording.

4) The organization of multiple Results sections appear lengthy and redundant. They should be combined, compressed, and reorganized. For example, the last section on correlations with AVA seems to contain the same information as "immobilization alters the correlation structure of neural activity". The sections / subsections "Population code for locomotion" (line 193) and "Largely distinct sub-populations contain information for velocity and curvature" (line 256) can be better organized.

I also view AVAL and AVAR coupling more as a benchmarking tool to give the readers confidence that their method works in the non-immobilized setting instead of an interesting new finding as it seems to be portraited in the abstract. Combining these results with an expanded sections to describe their imaging processing pipeline may be a better organization solution.

5) I personally found that among all results from the model, the notion that the simplest linear model works the best is the most interesting. It would be interesting to hear the authors' thoughts on its implication of the *C. elegans* brain network on motor states and their transitions.

*Reviewer #2 (Recommendations for the authors):*

My enthusiasm is diminished by a series of major concerns that I believe should be possible to address:

1) An important and interesting claim in the paper is that different neurons have different "tunings" for behavior – for example, some neurons are associated with forward velocity fluctuations, while others are associated with forward/reverse transitions. However, this is not very well explored in the paper. Some example data are shown, but that's about it. I'd suggest characterizing the full range of possible tunings that neurons can display and showing how many neurons in each of their datasets display such tunings. This could be a major strength of the paper if it is clearly characterized and communicated.

2) If the tunings are indeed diverse/complex (i.e. not just linear relationships), I'd suggest trying to predict behavior from single neurons using non-linear decoders. What is the best performance that can be obtained from single neurons using these more complex decoders? (and how does it compare to population-level decoders).

3) While it is readily apparent that the regression models perform better when trained from the full set of neurons (compared to the "best single neurons"), the authors' interpretation that this is because different neurons have different tunings does not yet seem fully supported. My main concern is that there is substantial levels of noise in their GCaMP measurements and that training models from more neurons may simply overcome this noise (the authors actually show that SNR impacts their predictive power in Figure 3-S1). For example, suppose that there were 2 neurons with perfectly correlated ground-truth activity and that they were both perfectly correlated with a behavior. If the activity measurements from these neurons had uncorrelated noise (noise in one neuron was not correlated with noise in the second), then a classifier trained to predict behavior would perform better if both neurons were used. In this case, this would not be due to any difference in the underlying tunings of the neurons. Are such effects occurring here? It is possible that one way to estimate the impact of these types of effects would be to compare models trained on similar amounts of data (e.g. 10min of data from one neuron vs. 5min of data from two simultaneously correlated neurons) or something like that. Another possibility would be to record single neurons (not in a whole brain context) in order to obtain higher SNR recordings and compare classifiers trained on these single neurons to those trained on the full population. (This would require knowing some of the "best single neurons")

4) Related to the above point, models with more parameters almost always perform better. To determine whether the increased model performance justified the use of additional parameters, I'd suggest using AIC (Akaike Information Criterion) or BIC (Bayesian Information Criterion) formulations.

5) The Introduction does not properly introduce what is known about the neural circuitry that gives rise to locomotion in *C. elegans*. The roles of many neurons have been carefully characterized – it would be useful to introduce what is known about their "tunings" from previous work and whether the field already thinks that a population code for locomotion may exist (or not).

6) In Figure 1 -S1 the authors compare velocity in their datasets, as measured by eigenworm analysis vs. center of mass movement. While they are correlated, I was surprised by how frequently they disagree. Why do they disagree at times? Are there errors in one or both of these methods?

7) In Figure 5, I believe it would be important to only present exemplary data from timepoints in the testing datasets, not the training datasets (i.e. only present correlation coefficients for datapoints in testing data; and only show examples of neural activity and behavior from testing data). For example, it is hard to know whether the relationships in Figure 5C are meaningful or just represent overfitting of the model if they are from the training data. (If these are test data already, please just make this clear in figure legend)

8) It is not clear that analyzing the weights in Figure 5A is really all that informative with regards to the underlying roles of the neurons. The fact that the model can predict behavior in withheld data is highly informative, but the specific weights recovered are influenced by the regularization method used, whether a neuron's activity contains information redundant with some other neuron's activity, etc.

9) There are no across-animal summary data of the effects that the authors show in Figure 5. This is just exemplary data. Are these observations consistent across animals?

*Reviewer #3 (Recommendations for the authors):*

1) Abstract would benefit from a statement of the main conclusion and its significance.

2) It would be helpful to motivate the immobilization experiment by first describing the state of knowledge concerning neuronal dynamics in worms (rather than waiting until the discussion).

3) What is the meaning of the shading in Figure 1d,e and similar places in the paper?

4) For readers unfamiliar with the *C. elegans* nervous system, it would be useful to make clear what fraction of all head neurons is being recorded, and also what fraction of all neurons is being recorded.

5) It might be more appropriate to move the section on correcting for motion artifacts (pg. 7 [171-182ff]) earlier in the paper, where this correction is first used. Or, move it to Methods.

6) Subscript (i) in Equation 1 is misplaced on pg. 7.

7) For those unfamiliar with the Fano factor, it might be worth pointing out that in Equation 1, the variance (numerator) refers to the signal, not the noise.

8) pg. 15 [379…]. "Our measurements suggest that neural dynamics from immobilized animals may not entirely reﬂect the neural dynamics of locomotion." Consider rephrasing. This sentence is almost a tautology as it says "…neural dynamics in the absence of locomotion may not entirely reflect the dynamics in the presence of locomotion."

9) Line 104-5: please add Faumont et al., 2011.

10) Line 198: Do you mean "Figure 5a,b"?

11) Line 206-7: Is neuron #29 actually in Figure 5x?

12) Line 344-5: Can you unpack this statement?

13) Line 359-361: Give particular examples of some circuit in which this statement is true.

---

## [Author Response]

Essential Revisions:1) Please address concerns by Reviewers #1 and #2 about identifying eigenworms with velocity and curvature (detailed in Recommendations for the authors).

In response to reviewer feedback, we have replaced the eigenworm analysis with more familiar definitions of velocity and curvature. Velocity is now the velocity of a point on the worm’s head in mm/s. Curvature is now the mean curvature of the centerline in κ = 𝑑θ radians/bodylength. We have revised all figures, and recalculated all models using these definitions. The conclusions remain the same. We thank the reviewers for this feedback and hope that these metrics of behavior will be more straightforward and perhaps more relevant to readers.

2) Please address the questions with respect to tuning and noise that are raised by Reviewer #2 (detailed in Recommendations for the authors).

Based on feedback from Reviewer #1 and #2, we have performed a new analysis and generated three new supplementary figures to characterize the full range of tunings and the number of tuned neurons in each dataset.

1. Figure 1 —figure supplement 1 shows the full range of F and dF/dT tunings with respect to velocity, including additional example tuning curves.

2. Figure 1 —figure supplement 2 shows the same for curvature.

3. Figure 1 —figure supplement 3 shows the number of significantly tuned neurons in each recording by type.

We respond to specific questions about noise in the more detailed response to Reviewer #2 further down. Briefly, AVA recordings shown in Figure 2 suggest that noise is not interfering with key features in our recordings. We thank the reviewer for posing an alternative hypothesis that multiple neurons may share the same ground truth signal and that population performance may merely reflect averaging over noisy copies of identical signals. In the new rewritten results section we now consider this explicitly and explain why our findings in Figure 5 suggest that it is unlikely the case. We have excerpted the relevant text in the detailed response to Reviewer #2.

3) Both Reviewer #1 and #3 (detailed in Recommendations for the authors) require that you address your conclusion that the population decoder outperforms the best single neuron. Is this a meaningful comparison, and how should such coding be interpreted?

We reorganized and revised the introduction, results and discussion to make three of our main points more clear:

1. The role of the population at representing locomotion has never before been explicitly tested in moving *C. elegans.*

2. Tuning of neurons across the population has not been systematically characterized in moving animals.

3. The combined result that the population performs better by leveraging diversity of tuning across the population constitutes a new and meaningful conclusion.

Reviewers #1 and #3 ask whether these conclusions are obvious or predictable. We argue they are not. We point to Reviewer #2 who wonders whether our findings might “not be due to any difference in the underlying tunings of the neurons” as further evidence that these findings are far from being a foregone conclusion. More details are in the individual responses to Reviewers #1 and #3.

4) The concerns of Reviewers #2 and #3 about the significance of the distribution of weights assigned by the decoder for how behavior is represented in the brain should be addressed (detailed in Recommendations for the authors).

We have rewritten that portion of the Results section to better motivate our analysis of neural weights and to clarify their significance. In particular, we now explicitly distinguish between aspects of the weights that are likely dictated by the choice of model and those aspects that are not penalized by the model. For example, the model could choose a different balance between positive and negative weights, or between weights assigned to neural activity and its temporal derivative, each without penalty. That these are roughly balanced, is more likely to reflect properties of behavior-related neural signals in the brain.

“To investigate how the decoder utilizes information from the population, we inspect the neural weights assigned by the decoder. The decoder assigns one weight for each neuron’s activity, W_F_, and another for the temporal derivative of its activity, 𝑊 _𝑑𝑓/𝑑𝑡_. It uses ridge regularization to penalize weights with large amplitudes, which is equivalent to a Bayesian estimation of the weights assuming a zero-mean Gaussian prior. In the exemplar recording from Figure 1, the distribution of weights for both velocity and curvature are indeed both well-approximated by a Gaussian distribution centered at zero. This suggests that the decoder does not need to deviate significantly from the prior in order to perform well. In particular, although changing the sign of any weight would not incur a regularization penalty, the decoder relies roughly equally on neurons that are positively and negatively tuned to velocity, and similarly for curvature. At the population level, the decoder assigns weights that are roughly distributed evenly between activity signals F and temporal derivative of activity signals dF /dt (Figure 5a,b). But at the level of individual neurons, the weight assigned to a neuron’s activity 𝑊_𝐹_ was not correlated with the weight assigned to the temporal derivative of its activity 𝑊 _𝑑𝑓/𝑑𝑡_ (Figure 5—figure supplement 1). Again, this is consistent with the model’s prior distribution of the weights. However, given that the model could have relied more heavily on either activity signals F or on temporal derivative signals dF /dt without penalty, we find it interesting that the decoder did not need to deviate from an even distribution of weights between them in order to perform well.”

5) All Reviewers (detailed in Recommendations for the authors) have strong suggestions for reorganizing the text and amplifying and deepening Introduction and Discussion. Reviewer #3's concerns about the functional implications of the decoding should be addressed. Limitations of the analysis should be clearly addressed in Discussion.

In response to this and other comments, we have rewritten the introduction and discussion We also explicitly include limitations of the analysis, for example:

“However, our results do not preclude the brain from using other methods for representing behavior. And in all cases, the measurements here do not distinguish between neural signals that drive locomotion, such as motor commands; and neural signals that monitor locomotion generated elsewhere, such as proprioceptive feedback (Wen et al., 2012). The decoder likely uses a mix of both. Future perturbation studies are needed to distinguish population-level signals that drive locomotion from those that monitor locomotion”

“And it is also possible that one of the non-linear models we did test would perform better with more training data. Complex models, including non-linear models, tend to have more parameters and are therefore prone to overfitting when trained on limited data. If a non-linear model performed poorly on our held-out data due to overfitting, it may perform better when trained with longer recordings. Poor performance here therefore does not inherently preclude a non-linear model from being useful for describing behavior signals in the *C. elegans* nervous system. Future work with longer recordings or the ability to aggregate training across multiple recordings is needed to better evaluate whether more complex models would outperform the simple linear decoder.”

“Future studies using newly developed methods for identifying neurons (Yemeni et al., 2020) are needed to reveal the identities of those neurons weighted by the decoder for decoding velocity, curvature, or both.”

We discuss these and other changes in more detail below.

Reviewer #1 (Recommendations for the authors):I hope that the authors focus on improving results and discussions sections of their strength (see above), including additional analyses, precise terminology, simplified statements, clarified discussions, and perhaps structural reorganization. I have a few concerns that I ask them to address or respond to, so that this work can be appreciated by and benefit the field. They are raised below, and should be viewed as suggestions for this purpose. (1) Line 71-85: This first Results section (which lacks a title) is a brief definition of the locomotion features for velocity and curvature as used throughout the paper.I am uncomfortable with the brevity of the introduction and justification of using eigenworms to represent velocity and curvature. These are two widely used biological terms, and the introduction would confuse many readers and even misled them (in the case of 'curvature').I share the authors' opinion on the deficiency of defining velocity by the animal's centroid displacement. However, they should be equally clear that their presentation for 'velocity' did not directly address this deficit: their analysis did not calculate and present the wave velocity – the speed of bending wave propagation – which would have the units of mm/sec or body lengths/sec as opposed to radians/sec.Moreover, in Figure 1-Figure S1, the authors demonstrated that their eigenvalue-derived velocity was well correlated with that of centroid-derived velocity values. This, to me, was a good validation to justify their choice of parameters as a proxy for velocity in later analyses. However, the authors did not cite this validation figure as its purpose, but instead in the context of a statement for the weakness of the centroid-based velocity measure. This is a misleading manipulation of citation of the authors' results.I have a bigger concern for referencing the third eigenworm as the 'curvature', specifically Lines 82-84 ("Here we report body curvature as a dimensionless quantity that captures bending in the dorsoventral plane, calculated by projecting the animal's body posture onto the third principal component of the eigenvalue decomposition."). To my understanding, this component best represents the body postures during turning. Their relationship with 'curvature' – which most would interpret not as a dimensionless quantity but as a precise measure of the degree of body bending per unit length – should be demonstrated similar to how the authors did so for velocity in Figure 1, Supplementary Figure 1. I personally consider it inappropriate to use 'curvature' when referring to the projections of the third eigenworm.

Based on this feedback, and on feedback from the other reviewers, we now use more familiar definitions of velocity and curvature. We now report velocity based on the movement of a point on the worm’s head in mm/s, and we report curvature as kappa or dtheta/dt in units of radians/bodylength. We updated all figures, recalculated all models, and rewrote the relevant methods sections. Many of the specific numerical values have changed, but the conclusions remain the same.

“To measure the animal's velocity we first find the velocity vector 𝑣 that describes the motion of a point on the animal's centerline 15% of its body length back from the tip of its head. We then project this velocity vector onto a head direction vector of unit length.

The head direction is taken to be the direction between two points along the animal's centerline, 10% and 20% posterior of the tip of the head. To calculate this velocity, the centerline and stage position measurements were first Hampel filtered and then interpolated onto a common time axis of 200 Hz (the rate at which we query stage position). Velocity was then obtained by convolving the position with the derivative of a Gaussian with σ = 0. 5s.

To measure the animal's average curvature < κ > at each time point, we calculated the curvature 𝑑θ/𝑑𝑠at each of 100 segments along the worm's centerline, where 𝑠 refers to the arc length of the centerline. We then took the mean of the curvatures of the middle segments that span an anterior-posterior region from 15% to 80% along the animal's centerline. This region was chosen to exclude curvature from small nose deflections (sometimes referred to as foraging) and to exclude the curvature of the tip of the tail.”

2) I found their motion correction important, interesting, and potentially useful to the community. The authors should definitely highlight it and elaborate in the text as a separate section instead of putting it away in Methods and at the end of the following Results section (Line 125: Population decoder outperforms best single neuron – this long result section can definitely benefit from 'de-mixing'.)To me, it would be very helpful to show the example data for the authors' methods for motion correction, including the raw traces of GCaMP and RFP before and after they performed correction by their ICA analyses (e.g. I think that it did not work as well for AVAL in Figure 2b; knowing what the trace was like before the correction would help me to examine why). I also would be curious to know why these authors limited their ICA to give two components instead of collecting all components and subtracting the ones correlated with RFP. It would be good if authors treated the number of ICA components as a parameter and explored the choice of this parameter on the performance of motion correction. A discussion on systematic ways to estimate this parameter would also be very welcome.

We appreciate the reviewer’s interest. ICA is one of many approaches we have tried as we continue to search for an optimal motion correction algorithm. Because motion correction is not the focus of this paper, we hesitated to dive into a deeper exploration of ICA. Ultimately, we decided to remove the ICA approach and replace it with a simpler regression approach that is better motivated and is easier to justify. The new simpler approach is described in the methods and excerpted below. It is well motivated under the assumption of linear additive noise; is computationally efficient; has only one free parameter; and is entirely deterministic, unlike common implementations of ICA. We have updated all figures and models using the new motion correction algorithm and revised the methods section. While numeric values throughout the paper changed, none of our conclusions changed upon switching motion correction algorithms.

“We used the GCaMP fluorescence together with the RFP fluorescence to calculate a motion corrected fluorescence, 𝐹_𝑚𝑐_ used through the paper. Note sometimes the subscript _𝑚𝑐_ is omitted for brevity. Motion and deformation in the animal's head

introduce artifacts into the fluorescent time-series. We assume that these artifacts are common to both GCaMP and RFP fluorescence, up to a scale factor, because both experience the same motion. For example, if a neuron is compressed during a head bend, the density of both GCaMP and RFP should increase, causing an increase in the fluorescence in both time-series. We expect that the RFP time series is entirely dominated by artifacts because, in the absence of motion, the RFP fluorescent intensity

would be constant. If we further assume that motion artifacts are additive, then a simple correction follows naturally. To correct for motion in the GCaMP fluorescence 𝐺, we subtract off a scaled RFP fluorescence, 𝑅, Fmc=(G−αR)−<G−αR> where α is a scaling factor that is fit for each neuron so as to minimize ∑(G(t)−αR(t))2

The final motion corrected signal 𝐹_𝑚𝑐_ is mean-subtracted.”

We note we have also revised how we set color bars for visualizing heatmaps of neural activity, as described in the methods:

“When presenting heatmaps of calcium activity, we set the colormap to visualize the motion-corrected data with the original, uncorrected dynamic range. Recordings in which neurons contain little signal compared to motion artifact will appear dimmer, while recordings in which neurons contain signal with large dynamics compared to the motion artifact will appear brighter. The limits on the colormap are determined by the uncorrected green fluorescent timeseries, specifically the 99th percentile of of all neurons at all time points ±|G− <G>| in the recording.”

Here are requested, smoothed, AVA traces before and after motion correction with the revised motion correction algorithm (mean-subtracted). We attribute the lower quality of the AVAL trace to AVAL’s spatial location at the far end of the animal relative to the microscope objective, as mentioned in the text. The difference between the original (F) and motion corrected (F_mc_) signals are subtle, as we expect. Note that previous recordings of AVA in moving worms report the sum of AVAR and AVAL because they cannot resolve the two independently. We now also report the sum in new Figure 2 —figure supplement 1 to facilitate comparison to the literature.

3) Section 'Population decoder outperforms best single neuron' and Figure 3aHere I have trouble appreciating the significance of this comparison. Previous studies have shown that forward, backward, and turning are three separate motor motifs of *C. elegans* locomotion. It is possible that multiple neurons may participate in multiple motor behaviors, but it would be truly astonishing (to me at least) if a single neuron plays a dominating role of all motifs of locomotion. Given the state of the field, scientifically it would be much more meaningful to compare the performance of a population decoder to the combination of the four best single neurons e.g. the best for positive velocity, the best for negative velocity, the best for dorsal turning, and the best for ventral turning, instead of one single best neuron.

This comparison has never been done before, and it is important to establish. We have revised our framing and added text clarifying the significance of the comparison. From the introduction:

“Despite growing interest in the role of population dynamics in the worm, their dimensionality, and their relation to behavior (Costa et al., 2019; Linderman et al., 2019; Brennan et al., 2019; Fieseler et al., 2020) it is not known how locomotory related information contained at the population level compares to that contained at the level of single neurons.”

In particular, we are also interested in how those signals combine:

“There has not yet been a systematic exploration of the types and distribution of locomotor related signals present in the neural population during movement and their tunings. So for example, it is not known whether all forward related neurons exhibit duplicate neural signals or whether a variety of distinct signals are combined.”

While it would be interesting to explore the four best single neurons, we worried that this would add confusion and disrupt the flow. We hope with the new framing the reviewers and editors see value in comparing against a single neuron.

The authors could also make it clear to readers that due to the lack of knowledge of neuronal identity, as well as the fact that each recording was capturing ~2/3 of the total neuronal population, the best single neuron decoder in each recording was only 'relative' to the captured neuronal population, and likely differed per recording.

We now clarify in the text:

“Because the correspondence between neurons across animals is not known in these recordings, the identities of neurons used by the population decoder and that of the specific best single neuron may vary from recording to recording.”

4) The organization of multiple Results sections appear lengthy and redundant. They should be combined, compressed, and reorganized. For example, the last section on correlations with AVA seems to contain the same information as "immobilization alters the correlation structure of neural activity". The sections / subsections "Population code for locomotion" (line 193) and "Largely distinct sub-populations contain information for velocity and curvature" (line 256) can be better organized.

Based on this feedback we have removed many of the subsection headings, combined others, and streamlined the text in places.

I also view AVAL and AVAR coupling more as a benchmarking tool to give the readers confidence that their method works in the non-immobilized setting instead of an interesting new finding as it seems to be portraited in the abstract. Combining these results with an expanded sections to describe their imaging processing pipeline may be a better organization solution.

We agree that the AVA recordings serve a benchmarking role. We have revised the abstract Accordingly:

“To validate our measurements, we labeled neurons AVAL and AVAR and found that their activity exhibited expected transients during backward locomotion.”

and updated the text in the Results section:

“To validate our population recordings, we investigated the well-characterized neuron pair AVAL and AVAR.”

We have left Figure 2 in the main text because the AVA recordings are an important and useful validation of our measurements.

5) I personally found that among all results from the model, the notion that the simplest linear model works the best is the most interesting. It would be interesting to hear the authors' thoughts on its implication of the *C. elegans* brain network on motor states and their transitions.

We have now added two paragraphs to the discussion:

“How should we interpret the finding that the decoder is linear? It has been observed that even very non-linear neural systems can encode information linearly. For example, the vertebrate retina has many highly non-linear connections but a linear decoder

performs indistinguishably from an artificial neural network at decoding visual signals from populations of retinal ganglian cells (Warland et al., 1997). *C. elegans* may be another example, like the retina, of a non-linear system that represents information

linearly. The *C. elegans* nervous system, however, also contains known instances of connections that appear linear over a physiologically relevant range of activities (Liu eet al., 2009; Lindsay et al., 2011, Narayan et al., 2011). So, it is also possible that the linear representation of behavior in *C. elegans* reflects linear circuitry in the brain.

We note that our exploration of non-linear models was not exhaustive. Although we tested a selection of non-linear models at the single neuron Figure 3 – Figure Supplement 5 and population level Figure 3 —figure supplement 4, it is possible that a different non-linear model would perform better. And it is also possible that one of the non-linear models we did test would perform better with more training data. Complex models, including non-linear models, tend to have more parameters and are therefore prone to overfitting when trained on limited data. If a non-linear model performed poorly on our held-out data due to overfitting, it may perform better when trained with longer recordings. Poor performance here therefore does not inherently preclude a non-linear model from being useful for describing behavior signals in the *C. elegans* nervous system. Future work with longer recordings or the ability to aggregate training across multiple recordings is needed to better evaluate whether more complex models would outperform the simple linear decoder.”

Reviewer #2 (Recommendations for the authors):My enthusiasm is diminished by a series of major concerns that I believe should be possible to address:1) An important and interesting claim in the paper is that different neurons have different "tunings" for behavior – for example, some neurons are associated with forward velocity fluctuations, while others are associated with forward/reverse transitions. However, this is not very well explored in the paper. Some example data are shown, but that's about it. I'd suggest characterizing the full range of possible tunings that neurons can display and showing how many neurons in each of their datasets display such tunings. This could be a major strength of the paper if it is clearly characterized and communicated.

Based on this feedback, and feedback from Reviewer #1, we added three new supplementary figures to characterize the full range of tunings and the number of tuned neurons in each dataset.

1. Figure 1 —figure supplement 1 shows the full range of F and dF/dT tunings with respect to velocity, including example tuning curves selected from neurons with a range of Pearson’s correlation coefficients.

2. Figure 1 —figure supplement 2 shows the full range of tunings with respect to curvature.

3. Figure 1 —figure supplement 3 shows the number of significantly tuned neurons by category in each recording.

2) If the tunings are indeed diverse/complex (i.e. not just linear relationships), I'd suggest trying to predict behavior from single neurons using non-linear decoders. What is the best performance that can be obtained from single neurons using these more complex decoders? (and how does it compare to population-level decoders).

Thank you for this suggestion. We now added Figure 3 —figure supplement 5 to show how various non-linear single neuron models compare to the linear population model.

3) While it is readily apparent that the regression models perform better when trained from the full set of neurons (compared to the "best single neurons"), the authors' interpretation that this is because different neurons have different tunings does not yet seem fully supported. My main concern is that there is substantial levels of noise in their GCaMP measurements and that training models from more neurons may simply overcome this noise (the authors actually show that SNR impacts their predictive power in Figure 3-S1). For example, suppose that there were 2 neurons with perfectly correlated ground-truth activity and that they were both perfectly correlated with a behavior. If the activity measurements from these neurons had uncorrelated noise (noise in one neuron was not correlated with noise in the second), then a classifier trained to predict behavior would perform better if both neurons were used. In this case, this would not be due to any difference in the underlying tunings of the neurons. Are such effects occurring here? It is possible that one way to estimate the impact of these types of effects would be to compare models trained on similar amounts of data (e.g. 10min of data from one neuron vs. 5min of data from two simultaneously correlated neurons) or something like that. Another possibility would be to record single neurons (not in a whole brain context) in order to obtain higher SNR recordings and compare classifiers trained on these single neurons to those trained on the full population. (This would require knowing some of the "best single neurons")

Quality varies across recordings, and we mention this in the text. But the evidence in Figure 2 does not suggest substantial levels of noise. Prior recordings (Ben Arous et al., 2010; Shipley et al., 2014) report the sum of AVAL and AVAR together because they cannot resolve the individual neurons. We now also report the sum of AVAL and AVAR’s activity in Figure 2 - Supplementary Figure 1 to make it easier to compare noise levels to previous reports. By comparing this trace with previous reports we conclude that the noise in this recording is not dramatically larger than in prior reports and, crucially, the noise is small compared to the relevant features of interest.

"We also report the sum of the individual traces in Figure 2—figure supplement 1. The similarity we observe between activities of AVAL and AVAR, and the similarities between our recordings of AVA and those previously reported in the literature serves to validate our ability to simultaneously record neural activity accurately from across the brain. It also suggests that the noise in this recording is modest compared to the features of interest in AVA’s calcium transients."

The evidence in Figure 5 suggests that the population decoder does not derive its performance simply by averaging over noisy neurons with the same ground-truth signals. In Figure 5d highly weighted neuron #77 has activity peaks at certain ventral turns while highly weighted neuron #84 has activity peaks at a complimentary set. It is unlikely that these neurons have the same ground truth signals, especially because we know from Figure 2 that the noise in this recording is small compared to features of interest (and these are from the same recording). The simplest explanation is that the decoder uses a variety of different types of neural signals from the population to decode. We thank the reviewer for raising this hypothesis because it is interesting and important to explore, and we now use it to frame this portion of our rewritten Results section:

“We wondered what types of signals are combined by the decoder. For example, it is conceptually useful to consider a simple null hypothesis in which multiple neurons exhibit exact copies of the same behavior-related signal with varying levels of noise. In that case, the population decoder would outperform the best single neuron merely by summing over duplicate noisy signals. We inspected the activity traces of the top weighted neurons in our exemplar recording (Figure 5c,d). Some highly weighted neurons had activity traces that appeared visually similar to the animal’s locomotory trace for the duration of the recording (e.g.#80 for curvature) and other neurons had activity that might plausibly be noisy copies of each other (e.g. #12 and #60 for velocity). But other highly weighted neurons had activity traces that were distinct or only matched specific features of the locomotory behavior. For example negatively weighted neuron #59 exhibited distinct positive peaks during dorsal turns (green arrows), but did not consistently exhibit corresponding negative peaks during ventral turns. This is consistent with prior reports of neurons such as SMDD that are known to exhibit peaks during dorsal but not ventral head bends (Hendricks et al., 2012; Shen et al., 2016; Kaplan et al., 2020).

In the recording shown, we also find some neurons that have activity matched to only specific instances of a behavior motif. For example, the temporal derivative of the activity of neuron #84 contributes distinct peaks to ventral bends at approximately 105 s and 210 s, but not during similar ventral turns at other time points (Figure 5d, blue arrows). Conversely, highly weighted neuron #77 contributes sharp peaks corresponding to four other ventral bends (Figure 5d, red arrows) that are absent from neuron #84. Similarly (although perhaps less striking) for velocity, neurons #24 and #110 contribute peaks for one set of reversals (Figure 5c, red arrows), while neuron #44 contributes peaks to a complimentary set of two reversals (Figure 5c, blue arrows). Similarly in recording AML32_A, different neurons contribute peaks of activity corresponding to different sets of ventral or dorsal turns, Figure 5 —figure supplement 3. While we observed this effect in some recordings, it was not obviously present in every recording. From this inspection of highly weighted neurons, we conclude that in at least some recordings the decoder is not primarily averaging over duplicate signals. Instead the decoder sums together different types of neural signals, including those that capture only a certain feature of a behavior (e.g. dorsal turns or ventral turns, but not both) or that seemingly capture only certain instance of the same behavior motif (some reversals but not others)."

4) Related to the above point, models with more parameters almost always perform better. To determine whether the increased model performance justified the use of additional parameters, I'd suggest using AIC (Akaike Information Criterion) or BIC (Bayesian Information Criterion) formulations.

We address concerns about overfitting by evaluating the model on held-out data. On held-out data, increasing the number of parameters does not always cause a performance increase. For example, Figure 3 —figure supplement 4h shows that a decision tree actually performs the worst of all models tested even though it has the most parameters (more than twice as many as any other model, as detailed in Table 5). We now explain this more clearly in the text:

“Evaluating performance on held-out data mitigates potential concerns that performance gains merely reflect over-fitting. In the context of held-out data, models with more parameters, even those that are over-fit, will not inherently perform better.”

And also in another section:

“Note that while adding features is guaranteed to improve performance on the training set, performance on the held-out test set did not necessarily have to improve.”

And we discuss implications of the over-fitting problem in the rewritten Discussion section:

“Complex models, including non-linear models, tend to have more parameters and are therefore prone to overfitting when trained on limited data. If a non-linear model performed poorly on our held-out data due to overfitting, it may perform better when trained with longer recordings. Poor performance here therefore does not inherently preclude a non-linear model from being useful for describing behavior signals in the C. elegans nervous system. Future work with longer recordings or the ability to aggregate training across multiple recordings is needed to better evaluate whether more complex models would outperform the simple linear decoder.”

5) The Introduction does not properly introduce what is known about the neural circuitry that gives rise to locomotion in *C. elegans*. The roles of many neurons have been carefully characterized – it would be useful to introduce what is known about their "tunings" from previous work and whether the field already thinks that a population code for locomotion may exist (or not).

Based on this, and other feedback, we have expanded the introduction to discuss more about what is known:

“The known locomotory circuitry in *C. elegans* focuses on a collection of pre-motor neurons and interneurons, including AVA, AVE, AVB, AIB, AIZ, RIM, RIA, RIV, RIB and PVC that have many connec-tions amongst themselves and send signals to downstream motor neurons involved in locomotion such as the A- or B-type or SMD motor neurons (White et al., 1976; Chalfie et al., 1985; Zheng et al., 1999; Gray et al., 2005; Gordus et al., 2015; Wang et al., 2020). These neurons can be grouped into categories that are related to forward locomotion, backward locomotion or turns. For example, AVA, AIB, RIM are part of a backward locomotory circuit (Zheng et al., 1999; Pirri et al., 2009; Gordus et al., 2015). AVB and PVC are part of a forward locomotion circuit (Gray et al., 2005; Chalfie et al., 1985; Zheng et al., 1999; Li et al., 2011; Xu et al., 2018) and RIV, RIB and RIA are related to turns (Gray et al., 2005; Li et al., 2011; Wang et al., 2020; Hendricks et al., 2012). Much of what we know about these neurons comes from recordings or manipulations of either single neurons at a time, or a selection of neurons simultaneously using sparse promoters (Gray et al., 2005; Guo et al., 2009; Arous et al., 2010; Kawano et al., 2011; Piggott et al., 2011; Gao et al., 2018; Wang et al., 2020). Only recently has it been possible to record from large populations of neurons first in immobile (Schrödel et al., 2013; Prevedel et al., 2014; Kato et al., 2015) and then moving animals (Nguyen et al., 2016; Venkatachalam et al., 2016).”

In the results and Discussion section we also try to provide more context and background. Selected examples:

“This is consistent with neurons such as RIVL/R that are active during ventral turns (Wang et al., 2020) or the SMDDs or SMDVs that have activity peaks during either dorsal or ventral head bends respectively (Hendricks et al., 2012; Shen et al., 2016; Kaplan et al., 2020).”

“This is consistent with other reports, including recent work suggesting that turning and reverse circuits are largely distinct modules except for a select few neurons, such as RIB, which may be involved in both (Wang et al., 2020)”

“…the temporal derivative of activity of AIB has been shown to be elevated during those reversals that are followed by turns compared to those followed by forward locomotion (Wang et al., 2020).”

“This is consistent with prior reports that for some neurons, like AVA, it is the temporal derivative of activity that correlates with aspects of locomotion (Kato et al., 2015) while for other neurons, such as AIY, it is the activity itself (Luo et al., 2014).”

6) In Figure 1 -S1 the authors compare velocity in their datasets, as measured by eigenworm analysis vs. center of mass movement. While they are correlated, I was surprised by how frequently they disagree. Why do they disagree at times? Are there errors in one or both of these methods?

For simplicity we now report velocity based on a point on the animal’s head, as described in the methods. All figures have been regenerated to reflect this change. We discuss this change in more detail in response to similar concerns raised by Reviewer #1.

“To measure the animal's velocity we first find the velocity vector 𝑣 that describes the motion of a point on the animal's centerline 15% of its body length back from the tip of its head. We then project this velocity vector onto a head direction vector of unit length. The head direction is taken to be the direction between two points along the animal's centerline, 10% and 20% posterior of the tip of the head. To calculate this velocity, the centerline and stage position measurements were first Hampel filtered and then interpolated onto a common time axis of 200 Hz (the rate at which we query stage position). Velocity was then obtained by convolving the position with the derivative of a Gaussian with σ = 0. 5s.”

(7) In Figure 5, I believe it would be important to only present exemplary data from timepoints in the testing datasets, not the training datasets (i.e. only present correlation coefficients for datapoints in testing data; and only show examples of neural activity and behavior from testing data). For example, it is hard to know whether the relationships in Figure 5C are meaningful or just represent overfitting of the model if they are from the training data. (if these are test data already, please just make this clear in figure legend)

Thank you for the suggestion. We now show which portion of the recording is held-out by adding light green shading to the traces in Figure 5c,d (same as Figure 3a,c) and we clarify in the caption.

8) It is not clear that analyzing the weights in Figure 5A is really all that informative with regards to the underlying roles of the neurons. The fact that the model can predict behavior in withheld data is highly informative, but the specific weights recovered are influenced by the regularization method used, whether a neuron's activity contains information redundant with some other neuron's activity, etc.

We agree that some properties of the neural weights are predetermined by choice of regularization. But many properties are not. We now clarify explicitly which features of the neural weights are expected from model choice, and which are not penalized by the model and therefore may more likely reflect properties of the brain:

“To investigate how the decoder utilizes information from the population, we inspect the neural weights assigned by the decoder. The decoder assigns one weight for each neuron’s activity, 𝑊_𝐹 ,_ and another for the temporal derivative of its activity, 𝑊_𝑑𝑓/𝑑𝑡._ It uses ridge regularization to penalize weights with large amplitudes, which is equivalent to a Bayesian estimation of the weights assuming a zero-mean Gaussian prior. In the exemplar recording from Figure 1, the distribution of weights for both velocity and curvature are indeed both well-approximated by a Gaussian distribution centered at zero. This suggests that the decoder does not need to deviate significantly from the prior in order to perform well. In particular, although changing the sign of any weight would not incur a regularization penalty, the decoder relies roughly equally on neurons that are positively and negatively tuned to velocity, and similarly for curvature.

At the population level, the decoder assigns weights that are roughly distributed evenly between activity signals F and temporal derivative of activity signals dF /dt (Figure 5a,b). But at the level of individual neurons, the weight assigned to a neuron’s activity 𝑊_𝐹_ was not correlated with the weight assigned to the temporal derivative of its activity 𝑊_𝑑𝑓/𝑑𝑡_ (Figure 5—figure supplement 1). Again, this is consistent with the model’s prior distribution of the weights. However, given that the model could have relied more heavily on either activity signals F or on temporal derivative signals dF /dt without penalty, we find it interesting that the decoder did not need to deviate from an even distribution of weights between them in order to perform well.”

9) There are no across-animal summary data of the effects that the authors show in Figure 5. This is just exemplary data. Are these observations consistent across animals?

We do not know of a satisfactory quantitative method for summarizing the effect in Figure 5 across recordings. Instead we now added a new example from a different recording in figure 5 —figure supplement 3. And we now clarify explicitly in the text:

“While we observed this effect in some recordings, it was not obviously present in every recording.”

Reviewer #3 (Recommendations for the authors):1) Abstract would benefit from a statement of the main conclusion and its significance.

A major significance of this work is that it provides needed experimental measurements to inform and constrain the interpretation of population dynamics. There is a growing body of literature interpreting population dynamics of locomotion (Kato et al., 2015, Costa et al., 2019, Linderman et al., 2019, Fieseler et al., 2020), yet until now fundamental measurements have been missing, such as the information in the population vs the single neuron or the relevance of immobile dynamics to study locomotion. We have revised the abstract to allude to this.

“We investigated the neural representation of locomotion in the nematode *C. elegans* by recording population calcium activity during movement. We report that population activity more accurately decodes locomotion than any single neuron. Relevant signals are distributed across neurons with diverse tunings to locomotion. Two largely distinct

subpopulations are informative for decoding velocity and curvature, and different neurons' activities contribute features relevant for different aspects of a behavior or different instances of a behavioral motif. To validate our measurements, we labeled neurons AVAL and AVAR and found that their activity exhibited expected transients during backward locomotion. Finally, we compared population activity during movement and immobilization. Immobilization alters the correlation structure of neural activity and its dynamics. Some neurons positively correlated with AVA during movement become negatively correlated during immobilization and vice versa. This work provides needed experimental measurements that inform and constrain ongoing efforts to understand population dynamics underlying locomotion in *C. elegans*.”

And we have better introduced these points in the introduction, excerpt is included in response to next item, below.

2) It would be helpful to motivate the immobilization experiment by first describing the state of knowledge concerning neuronal dynamics in worms (rather than waiting until the discussion).

We now do this, as suggested:

"Interestingly, results from recordings in immobile animals suggest that population neural state space trajectories in a low dimensional space may encode global motor commands (Kato et al., 2015), but this has yet to be explored in moving animals. Despite growing interest in the role of population dynamics in the worm, their dimensionality, and their relation to behavior (Costa et al., 2019; Linderman et al., 2019; Brennan and Proekt, 2019; Fieseler et al., 2020) it is not known how locomotory related information contained at the population level compares to that contained at the level of single neurons. And importantly, current findings of population dynamics related to locomotion in *C. elegans* are from immobilized animals. While there are clear benefits in studying fictive locomotion (Ahrens et al., 2012; Briggman et al., 2005; Kato et al., 2015), it is not known for *C. elegans* how neural population dynamics during immobile fictive locomotion compare to population dynamics during actual movement."

3) What is the meaning of the shading in Figure 1d,e and similar places in the paper?

Shaded circles show individual fluorescent time points. We now clarify:

“Blue or orange shaded circles show neural activity at each time point during behavior.”

4) For readers unfamiliar with the *C. elegans* nervous system, it would be useful to make clear what fraction of all head neurons is being recorded, and also what fraction of all neurons is being recorded.

Changed:

“To investigate locomotory-related signals in the brain, we simultaneously recorded calcium activity from the majority of the 188 neurons in the head…”

5) It might be more appropriate to move the section on correcting for motion artifacts (pg. 7 [171-182ff]) earlier in the paper, where this correction is first used. Or, move it to Methods.

We ultimately decided to keep that discussion of motion correction in its current location because it is pertinent when thinking about issues of noise, for example as raised by Reviewer #2. We also cover the key points in the section of the methods entitled “Motion-correction.”

6) Subscript (i) in Equation 1 is misplaced on pg. 7.

Thanks. Fixed.

7) For those unfamiliar with the Fano factor, it might be worth pointing out that in Equation 1, the variance (numerator) refers to the signal, not the noise.

Thank you for the suggestion:

“Here the variance term is related to the signal in the recording.”

8) pg. 15 [379…]. "Our measurements suggest that neural dynamics from immobilized animals may not entirely reﬂect the neural dynamics of locomotion." Consider rephrasing. This sentence is almost a tautology as it says "…neural dynamics in the absence of locomotion may not entirely reflect the dynamics in the presence of locomotion."

We have rewritten that section, reworded that statement and added context:

“That *C. elegans* neural dynamics exhibit different correlation structure during movement than during immobilization has implications for neural representations of locomotion. For example, it is now common to use dimensionality reduction techniques like PCA to search for low-dimensional trajectories or manifolds that relate to behavior or decision making in animals undergoing movement (Churchland et al., 2012; Harvey et al., 2012; Shenoy et al., 2013) or in immobilized animals undergoing fictive locomotion (Briggman et al., 2005; Kato et al., 2015). PCA critically depends on the correlation structure to define its principal components. In *C. elegans*, the low-dimensional neural trajectories observed in immobilized animals undergoing fictive locomotion, and the underlying correlation structure that defines those trajectories, are being used to draw conclusions about neural dynamics of actual locomotion. Our measurements suggest that to obtain a more complete picture of *C. elegans* neural dynamics related to locomotion, it will be helpful to probe neural state space trajectories recorded during actual locomotion: both because the neural dynamics themselves may differ during immobilization, but also because the correlation structure observed in the network, and consequently the relevant principal components, change upon immobilization. These changes may be due to proprioception (Wen et al., 2012), or due to different internal states associated with fictive versus actual locomotion."

9) Line 104-5: please add Faumont et al., 2011.

Added.

10) Line 198: Do you mean "Figure 5a,b"?

Yes. Thank you. Fixed.

11) Line 206-7: Is neuron #29 actually in Figure 5x?

This should be Figure 5c. Fixed. Thanks.

12) Line 344-5: Can you unpack this statement?

We have clarified with an example:

“One possible explanation is that superficially similar behavioral features like turns may actually consist of different underlying behaviors. For example, seemingly similar turns, on closer inspection, can be further subdivided into distinct groups (Broekmans et al., 2016).”

And we have added a related example:

“The neural representation associated with a motif may also depend on its behavioral context, including the behaviors that follow or proceed it. For example, the temporal derivative of activity of AIB has been shown to be elevated during those reversals that

are followed by turns compared to those followed by for- ward locomotion (Wang et al., 2020). The population may contain a variety of such neurons, each tuned to only a specific context of a given behavior, which would give rise to the neurons used by the decoder that are seemingly tuned to some instances of a motif and not others. The granularity with which to classify behaviors and how to take into account context and behavioral hierarchies remains an active area of research in *C. elegans* (Liu et al., 2018; Kaplan et al., 2020) and in other model systems (Berman et al., 2016; Datta et al., 2019). Ultimately, finding distinct neural signals may help inform our understanding of distinct behavior states and vice versa."

13) Line 359-361: Give particular examples of some circuit in which this statement is true.

We now provide an example:

"For example, both polymodal nociceptive stimuli detected from ASH (Mellem et al., 2002) and anterior mechanosensory stimuli detected from soft touch neurons ALM and AVM (Wicks and Rankin, 1995) activate reversals through shared circuitry containing AVA, among other common neurons. It is possible that the neural activities we observe for different behavioral motifs reflect sensory signals that arrive through different sensory pathways to evoke a common downstream motor response."